# On the Stability and Robustness of Vision Transformers for Neurodegenerative Disease Classification

**Eloi Navet**[1] [ID]                                                ELOI.NAVET@U-BORDEAUX.FR
**Rémi Giraud**[2] [ID]                                            REMI.GIRAUD@U-BORDEAUX.FR
**Boris Mansencal**[1] [ID]                                  BORIS.MANSENCAL@U-BORDEAUX.FR
**Pierrick Coupé**[1] [ID]                                      PIERRICK.COUPE@U-BORDEAUX.FR

[1] *Université de Bordeaux, CNRS, Bordeaux INP, LaBRI, UMR 5800, F-33400 Talence, France*
[2] *Université de Bordeaux, CNRS, Bordeaux INP, IMS, UMR 5218, F-33400 Talence, France*

**Editors:** Accepted for publication at MIDL 2026

## Abstract

Vision Transformers (ViTs) have recently been explored for structural MRI classification, motivated by their ability to capture non-local image structure. However, in limited and heterogeneous clinical cohorts, their weak inductive biases and sensitivity to training conditions often lead to high-variance behaviour. While binary settings such as cognitively normal vs. dementia are widely reported and typically exhibit moderate variability, we show that this stability does not extend to differential diagnosis. When increasing task complexity (e.g., controls vs. Alzheimer's Disease vs. Frontotemporal Dementia), performance becomes sensitive to class imbalance and phenotype overlap, with greater variability driven by fewer samples per class, noisier labels, and increased inter-site heterogeneity.

In this study, we investigate a stabilization protocol combining data augmentation, architectural constraints, and optimization strategies on multi-site MRI datasets. We assess how model variance evolves with task complexity using patient-level paired bootstrapping, calibration analysis, paired significance tests, and estimates of the probability of false outperformance to obtain uncertainty-aware comparisons across models.

Our results highlight conditions under which Transformer-based classifiers can be consistently trained with limited neuroimaging data and illustrate that several performance gains disappear once stochastic variability is reported. These results emphasize that reliable differential diagnosis with ViTs requires both robust stabilization protocols to mitigate optimization noise and standardized uncertainty quantification beyond simple point-estimates.

**Keywords:** Vision Transformers, Neurodegenerative Disease, Differential Diagnosis, Stability, Reproducibility, Robustness, Uncertainty Quantification.

## 1. Introduction

Transformer-based deep learning architectures such as ViT (Dosovitskiy et al., 2021) and Swin (Liu et al., 2021) are increasingly applied to neurodegeneration classification from structural MRI alone, due to their ability to capture distributed atrophy patterns that extend beyond local receptive fields (Shamshad et al., 2023; Alamir et al., 2024). Yet, unlike Convolutional Neural Networks (CNNs), these models lack the spatial inductive biases inherent to visual data (e.g., locality and translation invariance) and are typically trained on large, homogeneous datasets, which are conditions rarely met in clinical neuroimaging (Matsoukas et al., 2021). Medical data specificities, including inherent noise, acquisition heterogeneity, and severe class imbalance, exacerbate the instability of unregularized

transformers. Combined with the lack of intrinsic spatial constraints, these factors make optimization brittle, with sharp loss landscapes and heightened sensitivity to initialization, data ordering, and hyperparameters (Chen et al., 2022; Park and Kim, 2022).

Such instability is a critical bottleneck for the field. While CNNs remain robust baselines, they lack the unified token representation required to seamlessly integrate imaging with non-spatial modalities, such as genomics or tabular data in a unified embedding space (Wang et al., 2022b; Bi et al., 2024). Reported gains in medical imaging often lie within stochastic variability once uncertainty is quantified (Bouthillier et al., 2021), and recent audits show that formal significance testing is uncommon in leading venues (Christodoulou et al., 2025). These issues are particularly acute in classification tasks compared to segmentation. While voxel-wise segmentation metrics benefit from high information density per image, allowing for precise performance estimates even with moderate sample sizes, image-level classification relies on sparse supervisory signals (one label per volume) (El Jurdi et al., 2025). Consequently, classification tasks require substantially larger cohorts to achieve comparable statistical precision (Varoquaux, 2018), making them susceptible to the stochastic variability inherent in clinical datasets where sample sizes are typically constrained.

In neurodegenerative disease classification, while Cognitively Normal (CN) vs. Alzheimer Disease (AD) settings mask this difficulty (Basaia et al., 2019; Wen et al., 2020), multiclass tasks involving behavioral variant Frontotemporal Dementia (bvFTD), semantic variant Primary Progressive Aphasia (svPPA), and nonfluent variant Primary Progressive Aphasia (nfvPPA) frequently reveal ranking inversions between architectures and unstable decision boundaries. This instability is exacerbated by the reliance on purely imaging-based diagnosis; for example differentiating bvFTD from AD is particularly challenging solely from MRI due to overlapping atrophy patterns in the anterior cingulate and frontoinsula (Perry et al., 2017). This creates a "grey zone" where data-hungry models struggle to identify decision boundaries without the guidance of clinical or neuropsychological scores.

This work examines the stability and robustness of Transformers under the specific challenges of differential diagnosis on MRI. We identify techniques preserving calibration and ranking consistency across random seeds and distribution shifts. To this end, we propose:

1. **A stability assessment across task complexities:** By contrasting a standard 3-class setup (detailed in Appendix E) with a granular 5-class differential diagnosis (CN/AD/bvFTD/svPPA/nfvPPA), we quantify how increased phenotypic overlap and class imbalance amplify instability in Transformers compared to CNNs.

2. **A benchmarking of stabilization strategies:** Using a representative hierarchical vision transformer as a reference, we conduct an extensive ablation study reviewing the impact of data-level (augmentation, sampling), architectural (initialization, regularization), and optimization strategies. We identify a specific protocol that allows to close the generalization gap in limited-data regimes without relying on pretraining.

3. **An uncertainty-aware evaluation protocol:** Moving beyond point-estimates, we employ patient-level paired bootstrapping, calibration analysis, and the probability of false outperformance (Christodoulou et al., 2025). This framework allows us to differentiate genuine signal from optimization noise, *i.e.* that standard evaluation reports gains that fail to reach statistical significance with proper uncertainty quantification.

Our findings delineate the conditions under which Transformers can be deployed reliably for neurodegenerative disease classification. As architectures like MedViT show that Transformer-based models can surpass strong CNN baselines, our stabilization principles offer a pathway to unlock this potential across medical ViTs, clarifying which performance differences persist once stochastic variability and distribution shifts are accounted for. The code and stabilization protocols are publicly available.[1]

## 2. Related Work

Transformers have emerged as interesting alternatives to CNNs in medical imaging, offering the ability to capture long-range dependencies and global image structure (Dosovitskiy et al., 2021; Liu et al., 2021). While hierarchical variants like Swin (Liu et al., 2021) and domain-specific adaptations (e.g., Hatamizadeh et al., 2022; Wald et al., 2025a) attempt to mitigate the quadratic cost of attention and introduce inductive biases, training remains notoriously fragile. Unlike CNNs, Transformers lack spatial priors, making them prone to overfitting and unstable optimization on the small, heterogeneous cohorts typical of clinical neuroimaging (He et al., 2023; Shen et al., 2023).

Dementia classification has progressed from hand-crafted features with SVMs (Klöppel et al., 2008) to deep 3D CNNs (Wen et al., 2020; Nguyen et al., 2023) and, recently, Transformers (Nguyen et al., 2024). Current literature reports high performance for binary AD detection (Alamir et al., 2024). However, differential diagnosis involving FTD subtypes remains challenging due to overlapping phenotypes (Wu et al., 2025), a setting where the lack of rigorous uncertainty quantification is critical. Recent audits suggest that many reported performance gains in medical imaging may be attributable to stochastic variability rather than genuine architectural improvements (Christodoulou et al., 2025). This issue is exacerbated in Transformers due to their sensitivity to initialization and hyperparameters (Chen et al., 2022), requiring multi-seed evaluation (Bouthillier et al., 2021; Del Pup et al., 2024).

Addressing this instability requires a holistic approach to regularization and evaluation. Standard stabilization strategies include heavy data augmentation (Zhang et al., 2018; Cardoso et al., 2022), architectural constraints (e.g., LayerScale, stable initialization) (Touvron et al., 2021; Kedia et al., 2024), and optimization techniques such as sharpness-aware minimization (SAM) or label smoothing (Foret et al., 2021; Müller et al., 2020). Furthermore, relying solely on accuracy metrics is insufficient for clinical reliability. Robust assessment requires analyzing calibration (Guo et al., 2017) and validating statistical significance through paired tests and bootstrapping (McNemar, 1947; Efron, 1979). Complementarily, we employ the Brier score (Brier, 1950) to specifically quantify prediction over-confidence, offering an assessment of probabilistic reliability distinct from discrimination ranking. Our work tends to unify these disparate components, benchmarking stabilization strategies specifically for the low-data, high-imbalance framework of dementia differential diagnosis.

---

1. https://github.com/EloiNavet/ViT-Stability-Neurodegeneration/

## 3. Method

### 3.1. Datasets

We construct an in-domain (ID) pool combining data from the Alzheimer's Disease Neuroimaging Initiative (ADNI) (Mueller et al., 2005) and the multi-site ALLFTD consortium (Boeve et al., 2020), two longitudinal studies monitoring Alzheimer's Disease (AD) and Frontotemporal Lobar Degeneration (FTLD) respectively. We focus on the clinical spectrum of FTD subtypes, categorized into behavioral variant Frontotemporal Dementia (bvFTD), semantic variant Primary Progressive Aphasia (svPPA), and nonfluent variant Primary Progressive Aphasia (nfvPPA). To assess robustness to domain shifts, we employ an Out-Of-Domain (OOD) pool aggregating NIFD (Frontotemporal Lobar Degeneration Neuroimaging Initiative, 2009) and NACC (Beekly et al., 2007). Table 1 summarizes the class composition across these cohorts, while full cohort-level criteria and demographics are provided in Appendix A. Although this aggregated dataset is substantial by medical standards, it remains orders of magnitude smaller than the massive datasets typically required to train vision transformers in general computer vision.

**Distribution shifts and class imbalance.** While the ID cohort (ADNI+ALLFTD) already displays inherent class imbalance, it retains actionable representation for all FTD subtypes (see Table 1). In contrast, the OOD cohort (NIFD+NACC) shows an even more pronounced disparity, with rare FTD subtypes representing $< 1.5\%$ of the test sample. We preserve this natural prevalence to assess robustness under realistic epidemiological and covariate shifts. Evaluation is performed on unmodified OOD data; only ID training may use balanced sampling (Section 4.2.1). Due to scarcity, minority-class OOD metrics show wider confidence intervals and are interpreted primarily for global stability and calibration.

Table 1: **Subject distribution across cohorts.** The ID set combines ADNI and ALLFTD, and the OOD set aggregates NIFD and NACC. Note the scarcity of FTD subtypes in the OOD set reflecting clinical prevalence. Counts displayed as *Dataset1 / Dataset2* for each group.

| Group | Datasets | CN | AD | bvFTD | nfvPPA | svPPA | Total |
|---|---|---|---|---|---|---|---|
| ID | ADNI / ALLFTD | 1090 / 322 | 649 / 5 | – / 229 | – / 66 | – / 76 | 1739 / 698 |
| OOD | NIFD / NACC | 136 / 2115 | – / 485 | 74 / 26 | 37 / 6 | 39 / 4 | 286 / 2636 |

**Cross-validation and evaluation.** ID experiments use patient-level stratified 10-fold cross-validation across dataset, diagnosis, sex, and age bins, with a 7/2/1 train/val/test split per fold. OOD evaluation uses a fixed NIFD+NACC test set without overlap.

**Image preprocessing.** All T1-weighted MRIs undergo a unified pipeline: N4 bias correction (Tustison and et al., 2010), skull stripping, affine and diffeomorphic MNI registration (Avants et al., 2011; Fonov et al., 2011), 1 mm resampling, $Z$-scoring, and fixed FOV cropping. To avoid leakage from longitudinal data, a single baseline scan is selected per subject; CN participants are required to maintain longitudinal diagnostic stability. Genetic FTD cases in ALLFTD are excluded.

### 3.2. Models

**Backbones.** We benchmark a set of volumetric transformers and CNN baselines, selecting the variant model whose parameter count is close to the median across backbones:

1. **ViT-3D**: a non-hierarchical ViT extended to 3D via volumetric patch embedding (Dosovitskiy et al., 2021; Wang et al., 2022a);
2. **Swin-3D (baseline)**: a hierarchical Swin with shifted 3D windows (Liu et al., 2022);
3. **Swin-3D (deformable)**: Swin-3D equipped with deformable patch locations (Nguyen et al., 2024);
4. **MedViT-3D**: A robust hybrid CNN-Transformer model (Manzari et al., 2023), that we extended to 3D, combining local convolutions and global attention mechanisms;
5. **Segmentation-based CNNs + SVM**: A neuroanatomically driven pipeline using an ensemble of 125 3D U-Nets (AssemblyNet) (Coupé et al., 2020) for regional feature extraction, followed by an SVM classifier. This baseline serves as a high-capacity reference for anatomy-driven performance rather than a resource-equivalent competitor;
6. **3D CNN**: A 3D ResNet-18 (Hara et al., 2018). We selected the 18-layer variant to match the parameter count of the transformer architectures ($\approx$ 30M). While deeper variants theoretically offer higher capacity, 3D convolutions induce a rapid growth in parameter count, so the 18-layer variant offers a more favorable trade-off between model complexity and the available dataset scale.

**Rationale for stabilization testbed.** While hybrid architectures like MedViT-3D demonstrate superior baseline performance (as shown in Section 4), we explicitly selected Swin-3D with deformable patch location as the primary testbed for our stabilization ablation study (Section 4.2.1). Unlike MedViT, which relies on convolutional stems for stability, Swin-DPL allows us to isolate the optimization challenges intrinsic to hierarchical self-attention mechanisms. Our goal is to identify training protocols that allow standard Transformers to close the gap with hybrids and CNNs.

**Training framework.** Training uses AdamW (Loshchilov and Hutter, 2019), cross-entropy loss, global batch size 128 (with gradient accumulation), cosine decay with warmup (Loshchilov and Hutter, 2017), mixed precision, and early stopping on validation loss. We adopted this specific scheduler configuration as it represents the established standard for Vision Transformers (Dosovitskiy et al., 2021; Liu et al., 2021). While other optimization heuristics exist (e.g., gradient clipping, alternative schedules), we fixed these hyperparameters to focus our analysis on regularization and landscape smoothing techniques rather than exhaustive optimizer tuning. Regularization via Stochastic Depth (DropPath, (Huang et al., 2016)) and weight decay follows the specific configurations recommended by the respective model authors. All hyperparameters were fixed prior to experiments to isolate seed-dependent variation. While deterministic flags were enabled, strictly reproducible training remains elusive in 3D deep learning due to hardware-level implementation details.[2]

To rigorously quantify this instability, we employed a multi-seed evaluation protocol, repeating training runs with distinct random seeds for initialization and sampling. While

---

2. Despite deterministic seeds, atomic operations in specific 3D CUDA kernels (e.g., `avg_pool3d_backward_cuda`, `grid_sampler_3d_backward_cuda`) introduce irreducible bit-wise noise. This implementation-induced variability necessitates the multi-seed protocol described in Section 3.4.1.

this strategy is computationally expensive, multiplying the training budget by the number of seeds, it is strictly necessary to dissociate genuine architectural improvements from stochastic optimization noise when working with small datasets as we are. Unless stated otherwise, we ran 5 trainings per architecture for this analysis.

We evaluate stabilization components both individually and in combination. All models use the same preprocessed volumes, identical splits and compute budgets, except for SVM that requires training 125 U-Nets ($\approx$2.17M parameters each) (Coupé et al., 2020).

For each architecture, 10 models are trained (one per fold). ID performance is computed by concatenating the fold-specific test predictions. For OOD evaluation, we average the softmax outputs of the 10 fold-specific models.

### 3.3. Stabilization strategies

To mitigate the instability inherent to training Transformers on limited, heterogeneous MRI cohorts, we investigate a composite stabilization protocol. We categorize these strategies into data-centric, architectural, and optimization-based components. Mathematical formulations and implementation details are provided in Appendix C.

#### 3.3.1. TRAINING AND OPTIMIZATION

**Data regularization.** Given the high dimensionality of volumetric inputs relative to the cohort size, we employ extensive **data augmentation**. We use the MONAI framework (Cardoso et al., 2022) to apply domain-specific 3D transformations, including affine and elastic deformations, sagittal flips, bias-field simulation, and $k$-space artifact injection (see Appendix B for parameter ranges). To counteract decision boundary collapse in high-dimensional space, we employ **MixUp** (Zhang et al., 2018), which trains the network on convex combinations of sample pairs and their labels. This encourages the model to behave linearly in-between training examples. Furthermore, to address the severe class imbalance in the OOD settings (see Table 1), we use **balanced sampling**, ensuring that minibatches contain a uniform distribution of classes.

**Optimization landscape smoothing.** Standard Stochastic Gradient Descent (SGD) often converges to sharp minima in Transformers, which generalizes poorly. To try to mitigate this, we use **Sharpness-Aware Minimization (SAM)** (Foret et al., 2021), that simultaneously minimizes the loss value and the loss curvature, biasing the solution toward flatter regions of the loss landscape. We further smooth the optimization trajectory using **exponential moving average (EMA)** of model weights. By averaging parameters over the training trajectory, EMA provides a robust estimate of the "center" of the optimization basin, often yielding better generalization than the final checkpoint. Finally, we apply **label smoothing** to prevent the network from becoming over-confident on noisy labels, a critical factor given the phenotypic overlap in neurodegenerative diseases.

**Architectural constraints.** We evaluate signal propagation stabilization techniques: **LayerScale** (Touvron et al., 2021), which introduces learnable diagonal matrices to scale residual updates, and a **Stable Initialization** scheme (Kedia et al., 2024) to preserve activation variance throughout the network depth. We also benchmark **ShakeDrop** (Yamada et al., 2019), a stochastic regularization method within the residual block.

### 3.3.2. Inference and Evaluation

**Uncertainty-aware inference.** Comparison based on single point-estimates is unreliable when using small datasets. We therefore employ **checkpoint ensembling**, averaging the softmax predictions of the top-$K$ validation checkpoints. This acts as a simplified Snapshot Ensemble (Huang et al., 2017), marginalizing out local optimization noise. To address calibration, we apply post-hoc **temperature scaling** (Guo et al., 2017), optimizing a single scalar parameter on the validation set to align confidence scores with empirical accuracy. Finally, we explore **Test-Time Augmentation (TTA)**, aggregating predictions across multiple transformed views (flips, crops) of the test volume via inverse-entropy weighting, prioritizing views where the model is most confident.

### 3.4. Evaluation protocol

### 3.4.1. Instability quantification

To distinguish genuine architectural improvements from stochastic noise, we adopt a rigorous multi-seed evaluation protocol. Each fold is trained with a distinct seed, capturing variability arising from weight initialization, data ordering, and hardware nondeterminism.

We move beyond point-estimates by quantifying uncertainty via patient-level paired bootstrapping ($B = 10^4$ replicates) (Efron, 1979). While parametric assumptions can hold for dense segmentation tasks (El Jurdi et al., 2025), classification metrics on imbalanced cohorts often exhibit skewed, non-Gaussian distributions. We therefore opt for non-parametric bootstrapping to avoid distributional assumptions. We report the normalized Coefficient of Variation (nCV) to compare stability across datasets of varying sizes. This metric measures the variation between different trainings with the same configuration. The coefficient of variation (CV) is defined as the standard deviation divided by the mean metric, to integrate both the variability and the performance in one metric. It's normalized by the square root of the sample size to account for dataset size differences between classes and in-domain and out-of-domain, defining the normalized CV (see Appendix D for derivation). Statistical significance is assessed using a dual strategy: **paired Wilcoxon signed-rank tests** evaluate architectural stability across the 10 folds (in-domain), while **McNemar's test** (McNemar, 1947) assesses the diagnostic agreement of the final ensembled models (out-of-domain), with Bonferroni correction applied. Following Christodoulou et al. (2025), we also report the Probability of False Outperformance (PFO) to estimate the likelihood that a reported gain is not significant.

### 3.4.2. Metric Selection

**Discrimination metrics.** To address class imbalance (see Table 1), we report four complementary metrics: Accuracy (ACC), Matthews Correlation Coefficient (MCC), Macro-F1, and Precision-Recall AUC (PR-AUC). While **ACC** provides a standard overview, it is biased toward majority classes. Since Balanced Accuracy (BACC) can be inflated by saturated majority performance, we prioritize **MCC** as our primary ranking metric, that leverages the full confusion matrix and provide a robust correlation estimate regardless of class ratios. Complementarily, **Macro-F1** ensures equal contribution from all phenotypes, penalizing collapse on rare subtypes, while weighted one-vs-rest **PR-AUC** assesses discrim-

inatory power across decision thresholds, accounting for the varying support of each class. Finally, granular **per-class F1 scores** are detailed in Appendix E.

**Calibration.** Clinical deployment requires models to be trustworthy, not just accurate (Begoli et al., 2019). Standard discrimination metrics do not distinguish between a model that is cautiously wrong and one that is confident but wrong. We therefore evaluate reliability using the **Expected Calibration Error (ECE)** (Guo et al., 2017), which measures the alignment between predicted confidence and empirical accuracy (*i.e.*, "does a 90% confidence prediction imply a 90% probability of correctness?"). Complementarily, we employ the **Brier Score** (Brier, 1950), a proper scoring rule that penalizes over-confident false predictions. This metric helps identify over-confident models that assign high probability to incorrect classes, a critical failure mode in differential diagnosis that accuracy can mask.

## 4. Results

### 4.1. Instability in differential diagnosis

While preliminary experiments on a standard 3-class setup (CN/AD/FTD) confirmed the viability of Swin-DPL (see Appendix E), the transition to the 5-class differential diagnosis (CN/AD/bvFTD/svPPA/nfvPPA) reveals critical stability bottlenecks (Table 2). The granular classification induces a performance drop, highlighting the challenge of distinguishing phenotypically similar FTD subtypes.

In ID, pure Transformers struggle due to data scarcity: ViT-3D exhibits optimization collapse, failing to disentangle minority classes. This confirms that global attention mechanisms require more data or priors to find stable decision boundaries. Conversely, SVM and ResNet-18 maintain robust discrimination. The introduction of the Deformable Patch Location (DPL) allows Swin-3D to recover significant performance, aligning its ID convergence with CNN baselines.

OOD evaluation reveals a hierarchy shift. The hybrid MedViT-3D achieves the highest generalization in global metrics (ACC, MCC, PR-AUC), suggesting strong robustness to acquisition shifts. However, its lower Macro-F1 compared to SVM and ResNet indicates that this global performance comes at the cost of minority classes. While hybrids generalize well on average, strong inductive biases (SVM/CNN) seem better equipped to preserve recall on rare phenotypes (nfvPPA/svPPA) under distribution shifts. Finally, ViT-3D's low ECE reflects under-confidence rather than true calibration, a known limitation of the metric where uniform predictions yield low error (Nixon et al., 2019).

Transitioning from the 3-class to the 5-class setting, we observe that the widths of the bootstrap confidence intervals do not differ significantly. This highlights a limitation of bootstrapping, which is inherently constrained by the fixed sample size $N$ rather than the intrinsic difficulty of the optimization landscape. Similarly, while global nCV values remain comparable, the granular per-class analysis (see Appendix E) reveals highly non-uniform stability: variance spikes for minority phenotypes (nfvPPA, svPPA) where nCV exceeds 0.1 (vs <0.01 for controls), yet this volatility is masked in global metrics by the robustness of majority classes. These results show that 1) global metrics alone cannot fully characterize model reliability, and 2) greater task complexity introduces subtype-specific stochastic variability, motivating the proposed stabilization study.

Table 2: **Baseline performance (5-class: CN/AD/bvFTD/nfvPPA/svPPA differential diagnosis).** Performance metrics for Convolutional (ResNet, SVM) and Transformer (ViT, Swin) architectures on the CN/AD/bvFTD/nfvPPA/svPPA task. Results are reported for in-domain (ID, 10-fold CV) and out-of-domain (OOD, 10 models average) settings. **Bold** indicates the best performance per column.
Values: Mean $\pm$ 95% CI ($B = 10^4$), all metrics in %. $\uparrow$=higher-is-better, $\downarrow$=lower-is-better.

| | Configuration | Params | ACC $\uparrow$ | MCC $\uparrow$ | PR-AUC $\uparrow$ | Macro-F1 $\uparrow$ | ECE $\downarrow$ | Brier $\downarrow$ |
|---|---|---|---|---|---|---|---|---|
| **ID** | CNNs + SVM | $\approx$270M$^*$ | **82.80** $\pm$ **1.52** | **69.27** $\pm$ **2.53** | **67.72** $\pm$ **3.67** | **62.61** $\pm$ **3.68** | 47.84 | 11.43 |
| | ResNet-18 3D | 33.16M | 79.86 $\pm$ 1.56 | 64.04 $\pm$ 2.60 | 60.73 $\pm$ 3.61 | 57.47 $\pm$ 3.52 | 42.89 | **11.20** |
| | MedViT 3D | 34.99M | 78.83 $\pm$ 1.62 | 62.39 $\pm$ 2.63 | 58.37 $\pm$ 3.41 | 54.14 $\pm$ 3.12 | 44.04 | 11.65 |
| | ViT-3D | 23.18M | 69.39 $\pm$ 1.82 | 44.21 $\pm$ 2.84 | 39.41 $\pm$ 2.05 | 33.56 $\pm$ 2.29 | **36.65**$^\dagger$ | 12.68 |
| | Swin-3D | 29.27M | 73.33 $\pm$ 1.74 | 51.53 $\pm$ 2.90 | 52.15 $\pm$ 3.06 | 46.96 $\pm$ 2.80 | 39.07 | 12.12 |
| | Swin-3D DPL | 41.02M | 78.75 $\pm$ 1.65 | 61.60 $\pm$ 2.74 | 57.46 $\pm$ 3.06 | 49.57 $\pm$ 2.86 | 43.76 | 11.67 |
| **OOD** | CNNs + SVM | $\approx$270M$^*$ | 88.38 $\pm$ 1.17 | 69.67 $\pm$ 2.88 | 72.97 $\pm$ 4.51 | **67.97** $\pm$ **4.62** | 53.28 | 11.05 |
| | ResNet-18 3D | 33.16M | 88.90 $\pm$ 1.15 | 71.03 $\pm$ 2.86 | 73.58 $\pm$ 4.51 | 65.07 $\pm$ 3.72 | 52.54 | **10.71** |
| | MedViT 3D | 34.99M | **89.31** $\pm$ **1.12** | **72.16** $\pm$ **2.77** | **75.67** $\pm$ **4.31** | 61.99 $\pm$ 3.62 | 55.08 | 11.26 |
| | ViT-3D | 23.18M | 80.43 $\pm$ 1.44 | 49.74 $\pm$ 3.14 | 53.14 $\pm$ 3.69 | 29.75 $\pm$ 0.77 | **47.60**$^\dagger$ | 11.96 |
| | Swin-3D | 29.27M | 83.04 $\pm$ 1.35 | 57.56 $\pm$ 3.16 | 63.72 $\pm$ 4.31 | 50.76 $\pm$ 3.90 | 49.07 | 11.62 |
| | Swin-3D DPL | 41.02M | 85.74 $\pm$ 1.23 | 63.86 $\pm$ 2.95 | 70.64 $\pm$ 4.84 | 56.18 $\pm$ 3.57 | 50.64 | 11.22 |

$^*$Includes the segmentation backbone. $^\dagger$Low ECE here reflects under-confidence, not calibration.

## 4.2. Effect of stabilization

### 4.2.1. Training

Table 3 reports ablations of training-time stabilization components on Swin-3D DPL for the 5-class task. Domain-specific 3D MRI augmentation is the only isolated component that consistently improves OOD discrimination. In contrast, EMA weights and label smoothing, when used alone, induce small changes in accuracy and MCC but systematically degrade ECE and Brier, indicating that they sharpen the decision function at the expense of probability reliability. The cumulative protocol (`DA+E+LS+BS+M`) narrows the generalization gap with the stabilized ResNet-18 baseline without relying on architectural changes. While this composite strategy increases the computational budget, we provide a detailed cost-benefit analysis in Appendix H, demonstrating that the induced overhead is justified by gains in OOD reliability (e.g., +15% Macro-F1 for both ID and OOD). Class-balanced sampling and MixUp have limited effect when applied in isolation but are required in the cumulative setting to maintain non-zero F1 for minority FTD subtypes and to prevent decision-boundary collapse on nfvPPA and svPPA.

Figure 1 analyzes inter-seed stability *via* the normalized coefficient of variation across the training configurations of Table 3. Data augmentation and EMA consistently improve stability (lower nCV) across all metrics and domains, suggesting they effectively smooth the optimization landscape. Conversely, MixUp increases ID variability, particularly for ECE, while preserving OOD stability. This suggests that local variability prevents the model from settling into sharp minima, effectively trading training precision for better generalization.

Figures 2(a) and 2(b) summarize the statistical validation among the six training protocols of Table 3. We employ Wilcoxon to confirm that performance gains are consistent across training folds (ID, Figure 2(a)), and McNemar to validate the diagnostic superiority

Table 3: **Training stabilization ablation (5-class).** Quantitative comparison of training strategies applied to the Swin-3D DPL backbone. The table reports discrimination (ACC, MCC, PR-AUC, Macro-F1) and calibration (ECE, Brier) metrics for in-domain (ID, 10-fold CV) and out-of-domain (OOD, 10 models averaged predictions) settings. The upper section evaluates components applied individually to the baseline, while the lower section presents the results of the cumulative stabilization protocol.
Values: Mean $\pm$ 95% CI ($B = 10^4$), all metrics in %. ↑=higher-is-better, ↓=lower-is-better.

| | Configuration | ACC ↑ | MCC ↑ | PR-AUC ↑ | Macro-F1 ↑ | ECE ↓ | Brier ↓ |
|---|---|---|---|---|---|---|---|
| **ID** | Baseline (`BL`) | $78.75 \pm 1.65$ | $61.60 \pm 2.74$ | $57.46 \pm 3.06$ | $49.57 \pm 2.86$ | **43.76** | 11.67 |
| | + 3D Augments (`DA`) | $\mathbf{82.02} \pm \mathbf{1.52}$ | $\mathbf{68.12} \pm \mathbf{2.57}$ | $62.82 \pm 3.78$ | $59.26 \pm 3.68$ | 46.32 | **11.31** |
| | + EMA (`DA+E`) | $81.79 \pm 1.52$ | $67.42 \pm 2.56$ | $62.97 \pm 3.83$ | $61.15 \pm 3.72$ | 45.97 | 11.32 |
| | + Label smoothing (`DA+E+LS`) | $81.11 \pm 1.54$ | $66.15 \pm 2.60$ | $65.14 \pm 3.75$ | $58.88 \pm 3.83$ | 48.29 | 12.05 |
| | + Balanced sampling (`DA+E+LS+BS`) | $81.33 \pm 1.54$ | $66.89 \pm 2.62$ | $65.29 \pm 3.71$ | $61.42 \pm 3.62$ | 48.33 | 11.98 |
| | + MixUp (`DA+E+LS+BS+M`) | $81.62 \pm 1.54$ | $67.28 \pm 2.59$ | $\mathbf{66.43} \pm \mathbf{3.89}$ | $\mathbf{63.02} \pm \mathbf{3.71}$ | 48.94 | 12.04 |
| **OOD** | Baseline (`BL`) | $85.74 \pm 1.23$ | $63.86 \pm 2.95$ | $70.64 \pm 4.84$ | $56.18 \pm 3.57$ | **50.64** | 11.22 |
| | + 3D Augments (`DA`) | $87.39 \pm 1.22$ | $68.28 \pm 2.88$ | $72.72 \pm 4.91$ | $66.21 \pm 4.63$ | 51.61 | 10.94 |
| | + EMA (`DA+E`) | $88.24 \pm 1.20$ | $69.41 \pm 2.89$ | $74.64 \pm 4.52$ | $65.66 \pm 4.71$ | 52.22 | **10.83** |
| | + Label smoothing (`DA+E+LS`) | $88.34 \pm 1.16$ | $69.98 \pm 2.89$ | $74.10 \pm 4.81$ | $66.44 \pm 4.50$ | 55.25 | 11.65 |
| | + Balanced sampling (`DA+E+LS+BS`) | $88.14 \pm 1.19$ | $70.20 \pm 2.83$ | $\mathbf{75.30} \pm \mathbf{4.99}$ | $71.44 \pm 4.48$ | 55.06 | 11.65 |
| | + MixUp (`DA+E+LS+BS+M`) | $\mathbf{88.41} \pm \mathbf{1.19}$ | $\mathbf{70.38} \pm \mathbf{2.87}$ | $74.91 \pm 4.90$ | $\mathbf{71.54} \pm \mathbf{4.52}$ | 55.56 | 11.71 |

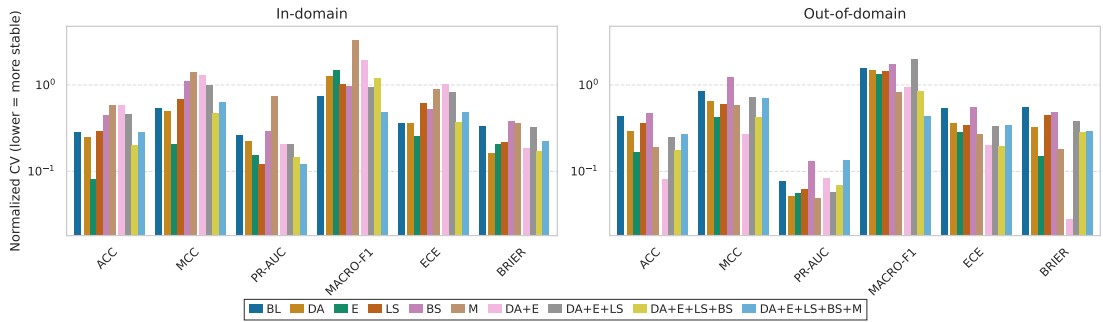

Figure 1: **Normalized coefficient of variation profiles of training strategies.** Bar plots displaying the inter-seed stability for 5 different seeds and multiple training stabilization components described in Table 3, covering both individual strategies and the cumulative protocol. Results are separated into in-domain (left) and out-of-domain (right) evaluations. We report accuracy, Matthews Correlation Coefficient, Precision-Recall AUC, Macro-F1, Expected Calibration Error, and Brier score for each individual component as well as the composition of them.

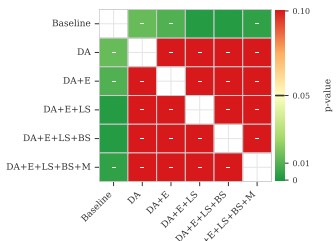
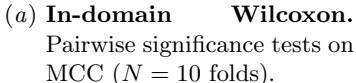
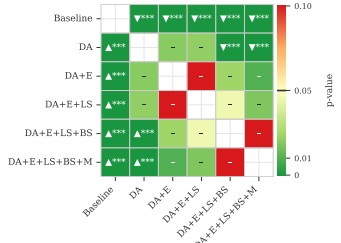
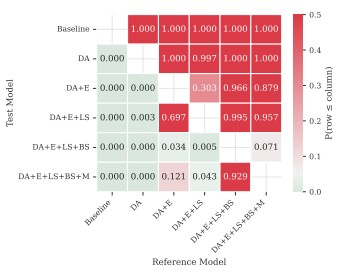

(a) **In-domain Wilcoxon.** Pairwise significance tests on MCC ($N = 10$ folds).

(b) **OOD McNemar.** Pairwise significance tests on aggregated predictions.

(c) **Reliability (PFO).** Probability of False Outperformance for OOD PR-AUC.

Figure 2: **Statistical assessment of stabilization and robustness.** Statistical validation matrices: (a) Wilcoxon signed-rank test $p$-values (in-domain), (b) McNemar test $p$-values (out-of-domain), and (c) Probability of False Outperformance (PFO) heatmap.

of the final ensembled system (OOD, Figure 2(b)). These matrices show that several numerical differences, including the improvement of the fully stabilized protocol over the baseline, reach standard significance thresholds, whereas intermediate variants such as DA+E often remain statistically indistinguishable despite visible shifts in mean performance. Beyond these training ablations, McNemar tests on OOD predictions also indicate that Swin-3D DPL significantly outperforms the standard Swin-3D backbone ($p < 0.05$), supporting the contribution of the deformable patch inductive bias. Complementarily, Figure 2(c) reports the PFO to assess the risk of illusory gains. It highlights that while the fully stabilized protocol yields a negligible PFO, intermediate strategies often exhibit high risk, *i.e.* their apparent gain may stem from stochastic variability rather than genuine signal.

### 4.2.2. MODELS

Table 4 benchmarks architectural regularization techniques commonly employed in large-scale vision transformers. Unlike the data-centric strategies identified in Section 4.2.1, architectural modifications proved ineffective or detrimental on this limited dataset.

Transitioning from Pre-Norm to Post-LN caused a collapse in discrimination, likely due to gradient vanishing in early stages, an issue typically mitigated by massive-batch pretraining not feasible with clinical MRI. Similarly, Sharpness-Aware Minimization (SAM), hypothesized to improve generalization, paradoxically degraded performance. This suggests that SAM's adversarial perturbations disrupt convergence when gradients are already noisy due to the restricted batch sizes of 3D training.

Regarding signal propagation, LayerScale (Touvron et al., 2021) proved hypersensitive: small initialization ($\gamma_0 = 10^{-5}$) prevented convergence, while $\gamma_0 = 0.1$ merely restored baseline parity. Passive regularization (ShakeDrop, Stable Initialization (Kedia et al., 2024)) offered no statistically significant improvement.

Crucially, the cumulative training protocol (bottom row) yields the only significant leap. This confirms that for medical ViTs trained from scratch, the bottleneck lies in the

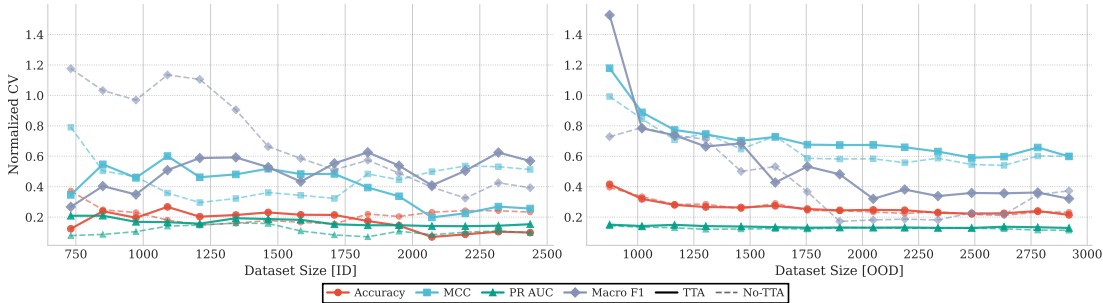

Figure 3: **Impact of sample size and Test-Time Augmentation (TTA) on stability.** Evolution of the normalized coefficient of variation across 5 random seeds as a function of dataset size. We compare standard inference (No-TTA, dashed lines) against TTA (solid lines) for both In-Domain (ID) and Out-of-Domain (OOD) settings. Note that while TTA acts as an effective stabilizer in the ID regime (lowering nCV, particularly for Macro-F1), this benefit does not consistently transfer to the OOD setting.

optimization landscape (smoothing *via* augmentation and averaging) rather than in the intrinsic architectural definition.

Table 4: **Model stabilization ablation (5-class). Architectural and optimization constraints** for in-domain (ID, 10-fold CV) and out-of-domain (OOD, 10 models averaged predictions). Standard vision strategies prove ineffective compared to the optimization protocol defined in Table 3.
Values: Mean $\pm$ 95% CI ($B = 10^4$), all metrics in %. $\uparrow$=higher-is-better, $\downarrow$=lower-is-better.

| | Configuration | ACC $\uparrow$ | MCC $\uparrow$ | PR-AUC $\uparrow$ | Macro-F1 $\uparrow$ | ECE $\downarrow$ | Brier $\downarrow$ |
|---|---|---|---|---|---|---|---|
| **ID** | Baseline | $78.75 \pm 1.65$ | $61.60 \pm 2.74$ | $57.46 \pm 3.06$ | $49.57 \pm 2.86$ | 43.76 | **11.67** |
| | Post-LN | $62.25 \pm 1.93$ | $25.94 \pm 2.71$ | $48.58 \pm 3.01$ | $29.92 \pm 3.34$ | **26.51** | $12.64^{\dagger}$ |
| | ShakeDrop | $\mathbf{78.83 \pm 1.62}$ | $\mathbf{62.28 \pm 2.65}$ | $58.27 \pm 3.37$ | $\mathbf{53.75 \pm 3.38}$ | 44.68 | 11.86 |
| | Stable Init | $77.88 \pm 1.64$ | $59.87 \pm 2.71$ | $\mathbf{58.31 \pm 3.32}$ | $48.83 \pm 2.88$ | 43.32 | 11.78 |
| | LayerScale ($\gamma_0$=0.1) | $78.17 \pm 1.62$ | $60.30 \pm 2.72$ | $57.61 \pm 3.33$ | $48.24 \pm 2.90$ | 43.70 | 11.79 |
| | SAM ($\rho$=0.05) | $66.68 \pm 1.87$ | $37.04 \pm 3.09$ | $39.76 \pm 2.48$ | $31.68 \pm 2.22$ | 34.62 | 12.99 |
| **OOD** | Baseline | $\mathbf{85.74 \pm 1.23}$ | $\mathbf{63.86 \pm 2.95}$ | $\mathbf{70.64 \pm 4.84}$ | $56.18 \pm 3.57$ | 50.64 | 11.22 |
| | Post-LN | $79.23 \pm 1.47$ | $28.48 \pm 3.36$ | $61.41 \pm 4.34$ | $36.60 \pm 4.62$ | **43.36** | $\mathbf{11.13}^{\dagger}$ |
| | ShakeDrop | $84.61 \pm 1.30$ | $62.67 \pm 2.97$ | $70.13 \pm 4.90$ | $\mathbf{58.35 \pm 3.80}$ | 50.48 | 11.53 |
| | Stable Init | $85.71 \pm 1.25$ | $63.66 \pm 2.99$ | $70.20 \pm 4.75$ | $56.82 \pm 3.49$ | 50.94 | 11.29 |
| | LayerScale ($\gamma_0$=0.1) | $85.50 \pm 1.27$ | $62.71 \pm 3.08$ | $68.71 \pm 4.79$ | $53.24 \pm 3.85$ | 50.78 | 11.30 |
| | SAM ($\rho$=0.05) | $77.97 \pm 1.51$ | $39.90 \pm 3.49$ | $47.18 \pm 3.64$ | $30.65 \pm 1.98$ | 46.14 | 12.33 |

$^{\dagger}$Low ECE and Brier here reflects under-confidence, not calibration.

### 4.2.3. Evaluation

To assess stability across imbalanced classes, we employ nCV to decouple intrinsic stability from sample size bias ($\sigma \propto 1/\sqrt{N}$) (see Appendix D.1). As shown in Figure 3, while raw CV decreases mechanically with $N$, nCV remains invariant once $N > 1000$, enabling fair comparison between majority classes and rare FTD subtypes.

Regarding inference-time strategies quantified in Table 5, snapshot ensembling emerges as the most effective method for variance reduction. By averaging predictions across the top-$K$ validation checkpoints, ensembling effectively marginalizes the local optimization noise inherent to the loss landscape of Transformers. As shown in Figure 11 (Appendix E), this strategy yields the lowest nCV across all metrics in both ID and OOD regimes. Furthermore, the stability analysis in Figure 4 indicates that these benefits saturate beyond $K = 12$ models, suggesting a diminishing return that balances computational cost with reliability. Complementarily, post-hoc temperature scaling significantly improves calibration, lowering the OOD Brier score without altering ranking metrics (PR-AUC), confirming that calibration errors can be addressed orthogonally to discrimination stability.

Table 5: **Inference-time strategies performance.** Quantitative comparison of Test-Time Augmentation (TTA), Temperature Scaling, and Snapshot Ensembling ($K = 12$) applied to the fully stabilized Swin-3D DPL model. We report discrimination (ACC, MCC, PR-AUC, Macro-F1) and reliability (ECE, Brier) metrics for in-domain and out-of-domain. Values: Mean $\pm$ 95% CI ($B = 10^4$), all metrics in %. ↑=higher-is-better, ↓=lower-is-better.

| | Variant | ACC ↑ | MCC ↑ | PR-AUC ↑ | Macro-F1 ↑ | ECE ↓ | Brier ↓ |
|---|---|---|---|---|---|---|---|
| **ID** | Baseline | $81.62 \pm 1.54$ | $67.28 \pm 2.59$ | $66.43 \pm 3.89$ | $63.02 \pm 3.71$ | 48.94 | 12.04 |
| | TTA | $\mathbf{82.07 \pm 1.54}$ | $\mathbf{68.82 \pm 2.51}$ | $\mathbf{67.75 \pm 3.83}$ | $66.73 \pm 3.38$ | 51.44 | 12.62 |
| | Temp. Scale | $81.54 \pm 1.52$ | $67.08 \pm 2.58$ | $65.61 \pm 3.98$ | $63.02 \pm 3.71$ | $\mathbf{46.24}$ | $\mathbf{11.44}$ |
| | TTA + Temp. | $\mathbf{82.07 \pm 1.54}$ | $\mathbf{68.82 \pm 2.51}$ | $67.17 \pm 3.97$ | $\mathbf{67.17 \pm 3.31}$ | 46.97 | 11.50 |
| | Ensemble | $82.02 \pm 1.54$ | $68.09 \pm 2.56$ | $65.31 \pm 3.79$ | $62.88 \pm 3.63$ | 49.78 | 12.15 |
| **OOD** | Baseline | $88.41 \pm 1.19$ | $70.38 \pm 2.87$ | $74.91 \pm 4.90$ | $71.54 \pm 4.52$ | 55.56 | 11.71 |
| | TTA | $86.43 \pm 1.23$ | $67.40 \pm 2.83$ | $74.66 \pm 5.00$ | $70.23 \pm 4.03$ | 55.85 | 12.44 |
| | Temp. Scale | $\mathbf{88.86 \pm 1.16}$ | $\mathbf{71.46 \pm 2.82}$ | $74.34 \pm 5.08$ | $71.54 \pm 4.50$ | 53.41 | $\mathbf{10.99}$ |
| | TTA + Temp. | $86.46 \pm 1.25$ | $67.63 \pm 2.83$ | $74.08 \pm 5.14$ | $70.67 \pm 3.99$ | $\mathbf{51.51}$ | 11.22 |
| | Ensemble | $88.31 \pm 1.18$ | $70.56 \pm 2.82$ | $\mathbf{75.90 \pm 4.62}$ | $\mathbf{72.15 \pm 4.40}$ | 55.85 | 11.83 |

Conversely, our results expose a key limitation of test-time augmentation (TTA) in neuroimaging. Although TTA is typically used to reduce variance, it proved detrimental in this differential-diagnosis setting, degrading OOD discrimination. As shown in Figure 3, TTA does not yield the consistent stability gains provided by ensembling. This suggests that the applied augmentations, though aligned with the training transforms, are either not representative of the OOD distribution or not well suited to our architecture.

## 5. Discussion

**Optimization landscape and the calibration-discrimination trade-off.** While ViTs theoretically offer global receptive fields capable of modeling distributed atrophy patterns (Dosovitskiy et al., 2021), their application to medical imaging is constrained by weak inductive biases and limited cohort sizes. Our findings suggest that the performance gap often observed between pure ViTs and CNNs stems largely from optimization: in small, heterogeneous clinical cohorts, sharp minima and sensitivity to initialization dominate model behaviour. Medical data specificities, such as acquisition noise, inter-site heterogeneity, and pronounced class imbalance, exacerbate this instability in unregularized Transformers.

Crucially, our analysis of the Brier score reveals a recurrent trade-off: models achieving the highest discrimination (Accuracy/MCC) often exhibit degradation in probabilistic reliability. Without regularization, the minimization of cross-entropy drives the network toward over-confident predictions (sharp decision boundaries); consequently, when these "superior" models are wrong, they do so with high confidence, penalizing the Brier score. We show that a targeted training scheme combining domain-specific 3D augmentation and regularization mitigates this behavior, enabling Swin-based models to match the robustness of strong baselines while preserving calibration in both in-domain and out-of-domain.

**The potential of Transformers and the hybrid bridge.** Contrary to the narrative that Transformers are inherently unsuitable for small-scale medical datasets, our results with MedViT-3D demonstrate that attention-based models can indeed outperform robust CNN baselines (ResNet-18) and segmentation ensembles. This validates the potential of token-based architectures to capture subtle, distributed markers of neurodegeneration that may elude purely local convolutional filters.

However, this performance must be considered carefully. The stability of MedViT stems largely from its hybrid design, where convolutional stems inject inductive biases that smooth the optimization. In contrast, pure or hierarchical Transformers, which lack this explicit structural guidance, exhibit high variance and optimization brittleness (as seen in Figure 7). While hybrids offer a performance gain, investigating the stabilization of standard Transformers remains essential. Unlike CNNs, whose performance tends to saturate, Transformers exhibit favorable scaling laws (Dosovitskiy et al., 2021) and offer the unified architecture required for future multimodal integration (e.g., fusing MRI, PET, and tabular neuropsychological scores). Establishing robust stabilization schemes for these backbones is therefore a prerequisite for deploying scalable, multimodal architectures in clinical settings without relying on convolutional backbones. While the integration of complementary modalities (e.g., MRI+PET+Tabular) is expected to reduce uncertainty by resolving phenotypic ambiguities, especially for FTD subtypes (Metz et al., 2025), it simultaneously increases input complexity. Just as we observed a counter-intuitive trade-off where improved discrimination came at the cost of calibration (increased Brier/ECE), we caution against the assumption that multimodal integration naturally guarantees stability. While new modalities introduce inductive biases, they also expand the token space, which may paradoxically exacerbate optimization instability. Therefore, rather than relying on intuition, we argue that the proposed stabilization principles remain important to prevent overfitting in these high-dimensional regimes.

**Disentangling data variance from optimization instability.** Our scaling analysis (Figure 5) aligns with theoretical expectations $(1/\sqrt{N})$: confidence interval widths decrease predictably as the number of subjects grows. As shown by El Jurdi et al. (2025), bootstrapping provides a computationally efficient way to estimate confidence intervals without distributional assumptions. However, it captures only data-driven variance (aleatoric uncertainty), not the intrinsic architecture instability (epistemic/optimization uncertainty). Thus, two models may exhibit identical bootstrap CIs yet differ greatly in their sensitivity to initialization. Ideally, architectural stability would be assessed by averaging predictions over many random seeds, but the cost of 3D training makes this impractical for routine development. Consequently, the stabilization techniques used here serve as a practical proxy:

by flattening the loss landscape and enforcing consistency, they reduce the irreducible error of single-seed training, which remains standard in clinical deep-learning deployment. It is important to distinguish between the computational cost of validating stability and the cost of deploying a stabilized model. While we employed extensive multi-seed evaluation to rigorously quantify optimization noise, this computationally intensive process serves as a research instrument rather than a requirement for deployment. The primary objective of the proposed stabilization protocol is to quantify and reduce inter-seed variability. By effectively reducing the nCV across architectures, the proposed protocol allows to rely on a single robust training cycle. This eliminates the need for prohibitive multi-seed ensembling in clinical practice, offering a favorable trade-off between training duration and diagnostic reliability (see Cost Analysis in Appendix H).

**Limitations.** The OOD cohort exhibits significant class imbalance, particularly for nfv-PPA and svPPA. Despite reporting macro-averaged metrics and employing balanced sampling, evaluation variance for these minority classes remains elevated, as confirmed by per-class stability analysis. Furthermore, label noise presents a specific challenge in the differential diagnosis of PPA. Clinical ground truth relies heavily on neuropsychological assessments and language tests, forming a multimodal composite reference standard. Training MRI-only models on these targets introduces inherent ambiguity, as structural signatures may lag behind or only partially reflect clinical phenotypes defined by non-imaging tests.

We acknowledge that this label noise is a significant factor. For example (Selvackadunco et al.) have shown that more than a third of clinical diagnoses for FTD present discordance between clinical diagnosis and final neuropathological diagnosis on their Brains for Dementia Research (BDR) cohort (Thomas, 2017). However, we argue that this constitutes an irreducible error (random uncertainty) inherent to large-scale clinical datasets (Karimi et al., 2020). Our results (Figures 8 and 9) demonstrate that while CNN baselines maintain low variance on these noisy classes, unregularized Transformers still exhibit volatility. This divergence proves that the instability we address is effectively optimization-driven, caused by the model overfitting to ambiguous labels, and is distinct from the data-inherent noise, which our protocol tries to mitigate.

Finally, regarding initialization strategies, we acknowledge the growing importance of self-supervised learning (SSL) on large-scale cohorts. However, our study explicitly targets the *from-scratch* training regime, which remains a standard scenario in clinical settings where massive unlabeled datasets or domain-specific foundation models are not always available due to privacy or data concerns. While SSL can inject strong inductive biases (Wald et al., 2025b), understanding how to stabilize architectures on limited data remains a prerequisite for their wider adoption. Future work should investigate the potential synergy between our optimization protocol and SSL initialization to further reduce variance in rare subtypes.

To summarize our findings into an actionable format, we provide a practitioner's guide in Table 6. Note that we do not provide any guidelines for architectural modifications (Section 4.2.2): even if these architectural changes do not yield good results in our case, they are subject to initialization conditions and the chosen architecture, which is why we cannot make a definitive conclusion.

Table 6: **Our guide to ViT stabilization in medical imaging.** Summary of best practices derived from our ablation study on differential diagnosis. We synthesize recommendations for Architecture, Training, Inference, and Evaluation to mitigate stochastic variability.

| Component | Recommendation | Rationale (Based on our findings) |
|---|---|---|
| **Architecture** | Prioritize Hierarchical (e.g., Swin) over Vanilla ViT | Vanilla ViTs exhibit optimization collapse on small 3D datasets; hierarchical biases stabilize convergence, see Table 11. |
| **Training** | Combine: 3D Aug. + EMA + MixUp + Balanced Sampling | Individual components are insufficient. The composite protocol closes the gap with CNNs and reduces variance, see Table 3 and Figure 1. |
| **Inference** | Snapshot Ensembling ($K \approx 12$); Investigate TTA applicability | Ensembling is the strongest variance reducer, see Figure 4. TTA may degrade OOD performance on asymmetric pathologies (e.g., PPA) as shown in Table 5. However, this behavior may be dataset-specific and highly dependent on the task and chosen augmentations; thus, this advice should be interpreted with caution. |
| **Evaluation** | Report nCV & Brier Score instead of pure Accuracy/ECE | Accuracy masks instability in minority classes, see Figure 9. ECE can be deceptively low due to under-confidence. |

## 6. Conclusion

This work examined the stability and reproducibility of Vision Transformers for neurodegenerative disease classification from structural MRI. All deep models exhibited non-negligible variability across seeds, with ViTs showing the highest sensitivity in the low-data, multiclass differential diagnosis setting involving FTD variants. Through systematic ablation, we showed that a tailored optimization protocol, combining domain-specific 3D MRI augmentation, optimization smoothing, and balanced sampling, substantially reduces variance and enables Swin-based models to approach the robustness of strong CNN baselines in both in-domain and out-of-domain evaluations, without modifying the backbone architecture.

Our uncertainty-aware evaluation framework, based on patient-level bootstrapping, calibration analysis, and the probability of false outperformance, revealed that differences that appear meaningful at the level of mean accuracy often fall within stochastic variability once uncertainty is quantified. Reproducible medical deep learning therefore requires going beyond single-seed point estimates to routinely report calibration, confidence intervals, and ranking stability across seeds and distribution shifts. Future work should investigate whether self-supervised pretraining on large unlabeled cohorts and the integration of non-imaging clinical covariates can provide additional inductive biases to stabilize Transformer-based classifiers while reducing the need for heavy regularization.

## Acknowledgments

This work benefited from the support of the project HoliBrain of the French National Research Agency (ANR-23-CE45-0020-01). This project is supported by the Precision and Global Vascular Brain Health Institute funded by the France 2030 investment plan as part of the IHU3 initiative (ANR-23-IAHU-0001). This study received financial support from the French government in the framework of the University of Bordeaux's France 2030 program / RRI "IMPACT" and the PEPR StratifyAging and PEPR Prodrom-ND. This work benefited from the support of the project ChatvolBrain from CNRS. This work was granted access to the HPC resources of IDRIS under the allocation AD011013926R3 made by GENCI.

The ADNI data used in the preparation of this manuscript were obtained from the Alzheimer's Disease Neuroimaging Initiative (ADNI) (National Institutes of Health Grant U01 AG024904). The ADNI is funded by the National Institute on Aging and the National Institute of Biomedical Imaging and Bioengineering and through generous contributions from the following: Abbott, AstraZeneca AB, Bayer Schering Pharma AG, Bristol-Myers Squibb, Eisai Global Clinical Development, Elan Corporation, Genentech, GE Healthcare, GlaxoSmithKline, Innogenetics NV, Johnson and Johnson, Eli Lilly and Co., Medpace Inc., Merck and Co. Inc., Novartis AG, Pfizer Inc., F. Hoffmann-La Roche, Schering-Plough, Synarc Inc., as well as nonprofit partners, the Alzheimer's Association and Alzheimer's Drug Discovery Foundation, with participation from the U.S. Food and Drug Administration. Private sector contributions to the ADNI are facilitated by the Foundation for the National Institutes of Health (www.fnih.org). The grantee organization is the Northern California Institute for Research and Education, and the study was coordinated by the Alzheimer's Disease Cooperative Study at the University of California, San Diego. ADNI data are disseminated by the Laboratory for NeuroImaging at the University of California, Los Angeles. This research was also supported by NIH grants P30 AG010129, K01 AG030514, and the Dana Foundation.

Frontotemporal Lobar Degeneration Neuroimaging Initiative (FTLDNI) was funded through the National Institute of Aging and started in 2010. The primary goals of FTLDNI was to identify neuroimaging modalities and methods of analysis for tracking frontotemporal lobar degeneration (FTLD) and to assess the value of imaging versus other biomarkers in diagnostic roles. The Principal Investigator of NIFD was Dr. Howard Rosen, MD, at the University of California, San Francisco. The data are the result of collaborative efforts at three sites in North America. For up-to-date information on participation and protocol, please visit http://memory.ucsf.edu/research/studies/nifd. Data collection and sharing for this project were funded by the Frontotemporal Lobar Degeneration Neuroimaging Initiative (National Institutes of Health). The study is coordinated through the University of California, San Francisco, Memory and Aging Center. FTLDNI data are disseminated by the Laboratory for Neuro Imaging at the University of Southern California.

The NACC database was funded by NIA/NIH Grants listed at https://naccdata.org/publishproject/authors-checklist#acknowledgment.

Data collection and dissemination of the data presented in this manuscript was supported by the ALLFTD Consortium (U19 AG063911, funded by the National Institute on Aging and the National Institute of Neurological Diseases and Stroke) and the former ARTFL and LEFFTDS Consortia (ARTFL: U54 NS092089, funded by the National Insti-

tute of Neurological Diseases and Stroke and National Center for Advancing Translational Sciences; LEFFTDS: U01 AG045390, funded by the National Institute on Aging and the National Institute of Neurological Diseases and Stroke). The authors acknowledge the invaluable contributions of the study participants and families as well as the assistance of the support staff at each of the participating sites.

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

## Appendix A. Extended dataset description

This appendix provides detailed inclusion criteria, cohort characteristics, and full demographic tables referenced in Section 3.1.

### A.1. Cohort overview

The In-Domain (ID) pool combines ADNI and ALLFTD, two harmonized longitudinal initiatives capturing AD and FTD clinical spectra. The Out-of-Domain (OOD) pool merges NIFD and NACC, which differ substantially in acquisition protocols, diagnostic granularity, recruitment strategies, and site diversity, thereby inducing both covariate and label distribution shifts.

### A.2. Subject selection

All analyses rely strictly on cross-sectional sampling. For each participant, we retain a single baseline T1w MRI to prevent leakage from repeated sessions, longitudinal progression, or variable scan quality. CN subjects must exhibit consistent diagnosis throughout follow-up. Participants with mixed or inconsistent diagnoses, atypical comorbidities, or missing metadata are excluded.

Because ALLFTD is genetically enriched, we remove carriers of known pathogenic mutations (e.g., *MAPT*, *C9orf72*, *GRN*) to align it with the predominantly sporadic composition of NIFD. For FTD subtyping, diagnostic labels follow each consortium's clinical adjudication protocols; ambiguous or overlapping classifications are excluded.

### A.3. Imaging pipeline

All T1w volumes undergo a unified preprocessing workflow to reduce site-specific variance:

1. N4 bias-field correction (Tustison and et al., 2010);

2. brain extraction;

3. affine + diffeomorphic ANTs registration to MNI (Avants et al., 2011; Fonov et al., 2011);

4. resampling to a 1 mm isotropic grid;

5. per-subject $Z$-score intensity normalization;

6. center-cropping to a fixed 3D field of view.

The same pipeline is applied across all cohorts. Scans failing QC after registration or skull stripping are discarded.

### A.4. Distribution shifts

The NIFD+NACC OOD pool exhibits pronounced long-tail distributions. FTD subtypes (especially svPPA and nfvPPA) are rare, accounting for fewer than 40 subjects per subtype. In contrast, CN and AD groups reach several thousand samples in NACC alone. This mismatch reflects real clinical prevalence rather than sampling artifacts.

These shifts imply:

- limited statistical power for subtype-resolved OOD metrics;

- sensitivity of bootstrap confidence intervals to minority-class scarcity;

- an increased role of calibration measures over raw accuracy;

- a stringent test of robustness to both covariate (scanner/site) and label (class ratio) shifts.

## A.5. Cross-validation strategy

ID experiments use 10-fold patient-level CV. Stratification jointly accounts for dataset membership, diagnostic category, biological sex, and discretized age (5-bin scheme). Within each fold, data are split into 70%/20%/10% train/validation/test. Each subject appears exactly once in an ID test set across folds.

OOD evaluation is performed on a fixed held-out NIFD+NACC set with no overlap with ID subjects.

## A.6. Detailed demographics and statistical assessment

Table 7: **Subject distribution across cohorts.** The in-domain (ID) set combines ADNI and ALLFTD, while the out-of-domain (OOD) set aggregates NIFD and NACC. Cells report the number of subjects with sex distribution *(Female/Male)* in the first line, and age *mean [min-max]* range in the second line. Dataset names include magnetic field strength and the count of unique scanner models.

| Group | Dataset | Diagnosis | | | | | Total |
|---|---|---|---|---|---|---|---|
| | | CN | AD | bvFTD | nfvPPA | svPPA | |
| ID | ADNI *(1.5T/3T)* *30 scanners* | 1090 (688/402) 69.7 [50-90] | 649 (281/368) 75.1 [55-94] | – | – | – | 1739 (969/770) 71.7 [50-94] |
| | ALLFTD *(3T)* *19 scanners* | 322 (203/119) 46.3 [18-79] | 5 (4/1) 66.8 [60-71] | 229 (76/153) 64.5 [40-85] | 66 (36/30) 68.9 [48-83] | 76 (39/37) 66.0 [50-86] | 698 (358/340) 56.7 [18-86] |
| | **Total** | **1412** (891/521) 64.4 [18-90] | **654** (285/369) 75.0 [55-94] | **229** (76/153) 64.5 [40-85] | **66** (36/30) 68.9 [48-83] | **76** (39/37) 66.0 [50-86] | **2437** (1327/1110) 67.4 [18-94] |
| OOD | NIFD *(3T)* *3 scanners* | 136 (77/59) 63.5 [39-81] | – | 74 (23/51) 61.8 [45-74] | 37 (20/17) 68.8 [54-81] | 39 (15/24) 63.4 [50-73] | 286 (135/151) 63.7 [39-81] |
| | NACC *(3T)* *16 scanners* | 2115 (1437/678) 68.1 [19-100] | 485 (266/219) 72.3 [38-96] | 26 (10/16) 64.4 [54-73] | 6 (4/2) 68.0 [57-77] | 4 (3/1) 64.9 [57-81] | 2636 (1720/916) 68.8 [19-100] |
| | **Total** | **2251** (1514/737) 67.8 [19-100] | **485** (266/219) 72.3 [38-96] | **100** (33/67) 62.5 [45-74] | **43** (24/19) 68.7 [54-81] | **43** (18/25) 63.5 [50-81] | **2922** (1855/1067) 68.3 [19-100] |

Table 7 details the repartition of diagnosis across datasets whereas Table 8 shows the demographic characteristics of the cohorts. We performed statistical testing to assess group differences. For age, a Kruskal-Wallis test revealed significant differences across diagnostic groups in both ID ($H = 421.28$, $p < 0.001$) and OOD ($H = 147.10$, $p < 0.001$) settings. For sex distribution, a Chi-square test also indicated significant variations in ID ($\chi^2 = 115.83$, $p < 0.001$) and OOD ($\chi^2 = 79.30$, $p < 0.001$).

We acknowledge that the cohorts are not explicitly matched. However, these differences largely reflect the epidemiological reality of the diseases (e.g., AD generally occurs later than subtypes of FTD). Although resampling matched subcohorts would eliminate these confounding factors, it would significantly reduce the sample size of already rare classes (e.g., nfvPPA/svPPA), making it even more difficult to train Vision Transformers, which require large amounts of data. Therefore, we prioritized maximizing sample size and diversity in order to study model stability under realistic

clinical conditions. Our stratified cross-validation strategy ensures that these demographic distributions are strictly identical between the training and validation sets within each fold to avoid bias during evaluation.

Table 8: **Detailed demographics and statistical assessment.** Distribution of age (Mean ± Std) and sex (Female/Male) across diagnostic groups. Statistical comparisons were performed using the Kruskal-Wallis test for age and Pearson's $\chi^2$ test for sex comparisons. Both tests reveal highly significant differences ($p < 0.001$) in In-Domain (ID) and Out-of-Domain (OOD) settings, statistically confirming that the groups are not demographically matched. This heterogeneity reflects the distinct epidemiological profiles of the diseases (e.g., AD typically has a later onset than FTD subtypes).

| Metric | Set | CN | AD | bvFTD | nfvPPA | svPPA | Stats (p-value) |
|---|---|---|---|---|---|---|---|
| **Age** | ID | $64.4 \pm 13.2$ ($n = 1412$) | $75.0 \pm 8.1$ ($n = 654$) | $64.5 \pm 7.8$ ($n = 229$) | $68.9 \pm 8.1$ ($n = 66$) | $66.0 \pm 7.2$ ($n = 76$) | $H = 421.3$ ($< 0.001$) |
| | OOD | $67.8 \pm 10.8$ ($n = 2251$) | $72.3 \pm 9.6$ ($n = 485$) | $62.5 \pm 6.2$ ($n = 100$) | $68.7 \pm 7.2$ ($n = 43$) | $63.5 \pm 6.6$ ($n = 43$) | $H = 147.1$ ($< 0.001$) |
| **Sex** (F/M) | ID | 891 / 521 (63% F) | 285 / 369 (44% F) | 76 / 153 (33% F) | 36 / 30 (55% F) | 39 / 37 (51% F) | $\chi^2 = 115.8$ ($< 0.001$) |
| | OOD | 1514 / 737 (67% F) | 266 / 219 (55% F) | 33 / 67 (33% F) | 24 / 19 (56% F) | 18 / 25 (42% F) | $\chi^2 = 79.3$ ($< 0.001$) |

# Appendix B. Augmentation protocols

Table 9 details the hyperparameter configurations for the MRI-specific data augmentation pipeline implemented via `MONAI`. All transformations are applied stochastically during training with the specified probabilities.

# Appendix C. Detailed stabilization protocols

This section provides the mathematical formulation and theoretical justification for the stabilization strategies employed in Section 3.3. We detail the specific hyperparameters used to ensure reproducibility.

## C.1. Data-Level regularization

**MixUp.** Standard empirical risk minimization often leads to memorization when $N$ is small. To counteract this, MixUp (Zhang et al., 2018) encourages the model to behave linearly between training examples. It generates synthetic samples $(x', y')$ by interpolating between random pairs of inputs $(x_i, y_i)$ and $(x_j, y_j)$:

$$x' = \lambda x_i + (1 - \lambda)x_j, \qquad y' = \lambda y_i + (1 - \lambda)y_j, \tag{1}$$

where the interpolation coefficient $\lambda$ is drawn from a Beta distribution $\lambda \sim \text{Beta}(\alpha, \alpha)$ with $\alpha = 0.3$. MixUp is applied after spatial and intensity transforms and preferentially pairs samples from **distinct classes** to enforce decision boundary regularization (Tokozume et al., 2018). In high-dimensional MRI space, this regularization prevents over-confident predictions in regions free of

Table 9: Detailed hyperparameters for 3D MRI augmentations. Probabilities ($p$) indicate the likelihood of applying the transform per sample. Note that sagittal flipping is included to enforce anatomical invariance during training.

| Category | Transform (MONAI) | Parameters |
|---|---|---|
| **Spatial** | RandAffine | $p = 0.5$, Rot $\pm30°$, Scale $\pm0.3$, Trans $\pm10$ vox, Padding: border |
| | Rand3DElastic | $p = 0.2$, $\sigma \in [5, 8]$, Magnitude $\in [100, 200]$ |
| | RandFlip | $p = 0.5$, Axis 0 (Sagittal) |
| **Intensity** | RandBiasField | $p = 0.3$, Coeff range $\in [0.0, 0.3]$ (Order 3) |
| | RandAdjustContrast | $p = 0.3$, $\gamma \in [0.7, 1.5]$ |
| | RandScaleIntensity | $p = 0.3$, Factor $\in [-0.5, 1.0]$ |
| | RandHistogramShift | $p = 0.2$, Control points $\in [5, 15]$ |
| | AdaptiveGaussianNoise | $p = 0.2$, Factor$= 0.1$ (injection relative to std) |
| | AdaptiveRicianNoise | $p = 0.2$, Standard Rician injection |
| **Artifacts** | RandGibbsNoise | $p = 0.2$, $\alpha \in [0.5, 1.0]$ |
| | RandKSpaceSpikeNoise | $p = 0.1$, Intensity $\in [13, 15]$ (k-space scale) |

training data, effectively smoothing the decision boundary between phenotypically similar classes, such as FTD subtypes.

Note that we also investigated CutMix (Yun et al., 2019) as an alternative mixing strategy. However, preliminary experiments yielded inferior performance compared to baseline or MixUp, likely because the rectangular region replacement destroys global anatomical context essential for analyzing distributed atrophy patterns. Consequently, we report only MixUp results.

**Class-aware balanced sampling.** Given the severe imbalance in the OOD cohort (see Table 1), standard uniform sampling would bias the gradient updates toward majority classes (CN, AD). We adjust the sampling probability $p_i$ for an image $x_i$ with label $y_i = c$ as:

$$p_i = \frac{1}{C \cdot n_c}, \tag{2}$$

where $C$ is the total number of classes and $n_c$ is the number of available samples for class $c$. This ensures that the expected number of samples per class in each mini-batch is uniform ($B/C$), preventing the minority FTD subtypes from being treated as outliers during optimization.

## C.2. Optimization dynamics

**Sharpness-Aware Minimization (SAM).** Vision Transformers trained on small datasets tend to converge to sharp local minima, which generalize poorly under distribution shifts. SAM (Foret et al., 2021) explicitly seeks parameters $w$ that lie in a "flat" neighborhood by solving a minimax game:

$$\min_w \mathcal{L}^{\text{SAM}}(w) \quad \text{where} \quad \mathcal{L}^{\text{SAM}}(w) = \max_{\|\epsilon\|_2 \leq \rho} \mathcal{L}(w + \epsilon). \tag{3}$$

Here, $\rho = 0.05$ is the radius of the perturbation neighborhood. In theory, by minimizing the loss under the worst-case weight perturbation $\epsilon$, SAM finds solutions robust to parameter noise. This

flatness is a proxy for generalization capability, essential when transferring models from ID to OOD domains.

**Label smoothing.** Medical diagnostic labels inevitably contain aleatoric uncertainty due to inter-rater variability. Training with "hard" one-hot targets $y_k \in \{0,1\}$ forces the model to be over-confident, often leading to overfitting. We relax the targets into soft probabilities $\tilde{y}_k$:

$$\tilde{y}_k = (1 - \varepsilon)y_k + \frac{\varepsilon}{K}, \tag{4}$$

where $K = 5$ is the number of classes and $\varepsilon = 0.1$ is the smoothing factor. This prevents the network from seeking infinite logit gaps for challenging samples (e.g., ambiguous early-stage dementia), resulting in better calibrated probabilities (lower ECE) as shown in Figure 3.

**Exponential Moving Average (EMA).** Stochastic Gradient Descent introduces noise into the optimization trajectory, particularly with small batch sizes. EMA maintains a "shadow" model with weights $\tilde{\theta}$ that are updated at each step $t$ using the current online weights $\theta_t$:

$$\tilde{\theta}_t = \frac{\sum_{k=0}^{K-1} \beta^k \theta_{t-k}}{\sum_{k=0}^{K-1} \beta^k}. \tag{5}$$

We use a slow decay rate $\beta = 0.999$ and $K = 3$ to average the last 3 model weights. This acts as an averaging filter, effectively smoothing out the high-frequency oscillations of the optimization path. It provides a more stable estimate of the central tendency of the loss basin.

### C.3. Architectural Constraints

**LayerScale.** Deep Transformers (like Swin) often suffer from signal degradation in deeper layers. LayerScale (Touvron et al., 2021) facilitates signal propagation by introducing a learnable diagonal matrix $\Lambda_l$ to scale the output of the residual block $\mathcal{F}$:

$$x_{l+1} = x_l + \Lambda_l \cdot \mathcal{F}(\text{LN}(x_l)), \quad \Lambda_l = \text{diag}(\lambda_{l,1}, \ldots, \lambda_{l,d}). \tag{6}$$

Initializing $\lambda$ to a small value (e.g., $10^{-5}$) allows the network to behave closer to an identity function at the start of training, easing the optimization of deep architectures on small datasets where gradients might otherwise vanish or explode.

### C.4. Inference-time aggregation

**Test-Time Augmentation (TTA) with entropy weighting.** We apply $M = 8$ transformations $T_m$ (flips, crops) to each test volume $x$. Simple averaging can be detrimental if certain views (e.g., occluded crops) yield noisy predictions. We therefore use inverse-entropy weighting to prioritize confident predictions. The final probability $\bar{y}$ is:

$$\bar{y} = \sum_{m=1}^{M} w_m p_m(T_m(x)), \quad w_m \propto \frac{1}{H(p_m) + \xi}, \tag{7}$$

where $H(p_m)$ is the entropy of the prediction and $\xi$ is a stability constant. This explicitly marginalizes over geometric nuisance variables, enforcing invariance to acquisition variations that the model may not have fully learned during training.

**Ensemble of snapshots.** To determine the optimal ensemble size $K$, we analyzed the evolution of inter-seed stability as a function of the number of aggregated checkpoints. Figure 4 illustrates the nCV for discrimination metrics across varying $K$. We observe a characteristic convex profile in the In-Domain setting: while increasing $K$ initially reduces variance by marginalizing out local optimization noise, the stability benefits saturate and even degrade beyond $K = 12$. This inflection point likely indicates that expanding the ensemble further necessitates the inclusion of suboptimal checkpoints (ranked 13th and below on validation data), which dilutes the consensus and re-introduces variance. Consequently, we selected $K = 12$ as the operating point for all reported experiments, providing the optimal trade-off between variance reduction and computational inference cost.

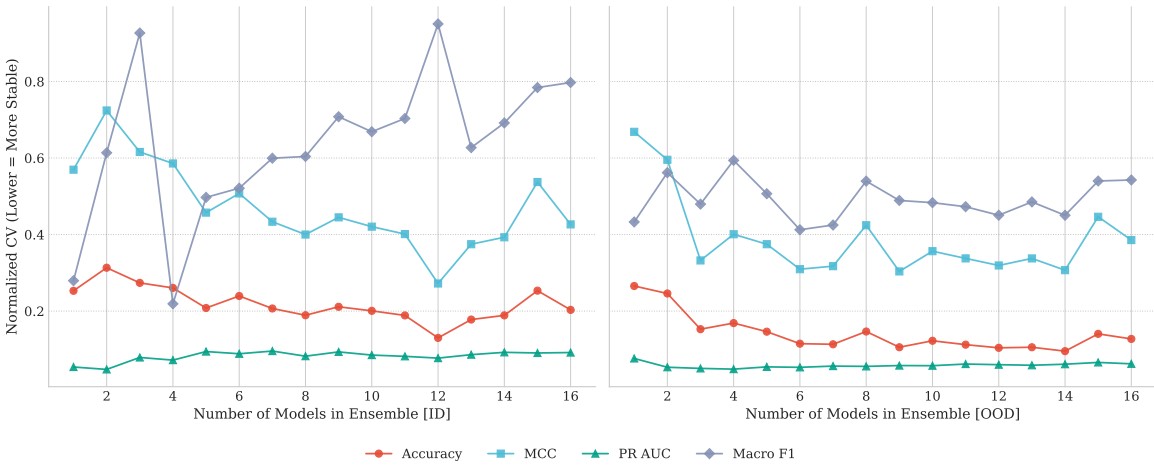

Figure 4: **Effect of ensemble size on stability.** Evolution of the normalized CV for Accuracy, MCC, and PR-AUC as a function of the number of models in the snapshot ensemble ($K$).

## Appendix D. Evaluation metrics and definitions

This appendix provides the formal definitions of the metrics used in the evaluation protocol.

### D.1. Stability metrics

**Normalized Coefficient of Variation (nCV).** Figure 5 validates uncertainty estimates by tracking 95% CI width against test set size. The widths follow a theoretical $C/\sqrt{N}$ decay ($R^2 > 0.9$ for Accuracy/MCC), confirming that reported instability is intrinsic to the models rather than a sampling artifact.

Standard deviation naturally decreases as sample size $N$ increases ($\sigma \propto 1/\sqrt{N}$). To compare the intrinsic stability of models across classes with vastly different sizes (e.g., 2000 CN vs. 40 nfvPPA), we use the nCV to decouple stability from sampling density:

$$\text{nCV} = \sqrt{N}\frac{\sigma}{\mu}, \tag{8}$$

where $\mu$ and $\sigma$ are the mean and standard deviation of the metric across seeds. This normalization allows for a fair comparison of variance between majority and minority classes.

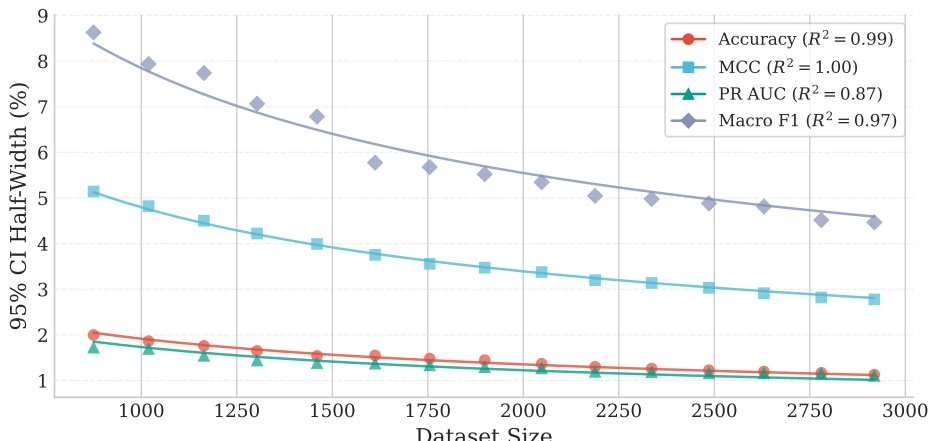

Figure 5: **Confidence Interval Scaling Analysis.** Evolution of the 95% CI half-width as a function of the training set size $N$ (subsampled from the OOD cohort). The solid lines represent the theoretical fit $y = C/\sqrt{N}$. The high $R^2$ values indicate that the estimated uncertainty strictly follows expected statistical laws, validating the reliability of the reported variance.

**Probability of False Outperformance (PFO).** The PFO estimates the probability that a baseline model $A$ is actually superior to a proposed model $B$, despite the observed mean difference $\bar{\delta} > 0$. It is computed directly from the bootstrap replicates:

$$\widehat{\Pr}(\Delta \leq 0) = \frac{1}{B} \sum_{b=1}^{B} \mathbb{1}(\delta_b \leq 0), \tag{9}$$

where $\delta_b$ is the performance difference in the $b$-th bootstrap sample.

### D.2. Classification Metrics

To ensure consistency and readability throughout the paper, all reported metrics (discrimination and reliability) are scaled by a factor of 100 and expressed as percentages.

To formally define the metrics, let $K$ be the $C \times C$ confusion matrix where $K_{ij}$ represents the number of samples of class $j$ predicted as class $i$. We define the total samples $s = \sum_{ij} K_{ij}$, the total correct predictions $c = \sum_k K_{kk}$, the total predictions for class $k$ as $p_k = \sum_j K_{kj}$, and the total true labels for class $k$ as $t_k = \sum_i K_{ik}$.

**Accuracy.** Standard accuracy measures the overall proportion of correct predictions. While intuitive, it can be misleading in imbalanced settings where majority classes dominate the score.

$$\text{ACC} = \frac{\sum_k K_{kk}}{s} = \frac{c}{s}. \tag{10}$$

**Matthews Correlation Coefficient (MCC).** We employ the multiclass generalization of the MCC. Unlike F1 or Accuracy, MCC involves all four quadrants of the confusion matrix (True Positives, False Positives, True Negatives, False Negatives), making it the most robust single-value metric for imbalanced datasets.

$$\text{MCC} = \frac{c \cdot s - \sum_k p_k t_k}{\sqrt{(s^2 - \sum_k p_k^2)(s^2 - \sum_k t_k^2)}}. \tag{11}$$

**Precision-Recall AUC.** The PR-AUC assesses the trade-off between Precision ($P$) and Recall ($R$) across different decision thresholds $\tau \in [0, 1]$. For a multi-class problem, we compute the Area Under the Curve (AUC) for each class $k$ in a one-vs-rest manner and report the macro-average (unweighted mean) to ensure equal contribution from all phenotypes regardless of prevalence:

$$\text{PR-AUC} = \frac{1}{C} \sum_{k=1}^{C} \int_0^1 P_k(R) \, dR, \tag{12}$$

where $P_k = \frac{K_{kk}}{p_k}$ and $R_k = \frac{K_{kk}}{t_k}$ are the precision and recall for class $k$ computed at varying operating points.

**Per-class F1-Score.** The F1-score for a specific class $k$ is the harmonic mean of its precision and recall. It effectively penalizes the model if it fails to retrieve instances of class $k$ (low recall) or hallucinates them (low precision).

$$\text{F1}_k = 2 \cdot \frac{P_k \cdot R_k}{P_k + R_k} = \frac{2K_{kk}}{p_k + t_k}. \tag{13}$$

**Macro F1-Score.** To obtain a global performance metric that treats all classes equally regardless of their support size (prevalence), we compute the unweighted mean of the per-class F1 scores.

$$\text{Macro-F1} = \frac{1}{C} \sum_{k=1}^{C} \text{F1}_k. \tag{14}$$

### D.3. Reliability Metrics

**Expected Calibration Error (ECE).** We calculate the ECE (Guo et al., 2017), which approximates the expected difference between the model's confidence and its actual accuracy. Following standard practice, we employ a fixed discretization scheme with $M = 15$ equidistant bins based on the maximum softmax probability:

$$\text{ECE} = \sum_{m=1}^{M} \frac{|B_m|}{N} \big| \text{acc}(B_m) - \text{conf}(B_m) \big|, \tag{15}$$

where $B_m$ is the set of samples in bin $m$, $\text{acc}(B_m)$ is the accuracy within the bin, and $\text{conf}(B_m)$ is the average confidence. Lower ECE indicates better calibration. In a clinical setting, a low ECE is critical as it implies that a prediction made with 90% confidence indeed corresponds to a 90% probability of correctness, fostering trust in the decision support system.

**Brier Score.** The Brier score (Brier, 1950) is computed as the Mean Squared Error between the predicted probability distribution and the one-hot encoded ground truth. To align its scale with accuracy, we report it as a percentage:

$$\text{Brier} = \frac{1}{N \cdot C} \sum_{i=1}^{N} \sum_{k=1}^{C} \big( P(y_i = k \mid x_i) - \mathbb{1}[y_i = k] \big)^2. \tag{16}$$

Here, the score is normalized by the number of classes $C$, preventing mechanical inflation due to task dimensionality. Unlike accuracy, which relies solely on the ranking, the Brier score heavily penalizes over-confident false predictions. It thus provides a holistic assessment of probabilistic reliability, favoring models that remain uncertain when evidence is ambiguous.

## Appendix E. Instabilities arise going from 3 to 5 classes

Table 10 summarizes the baseline performance on the standard 3-class task. As detailed in the Introduction, differentiating bvFTD from AD presents a significant challenge due to overlapping atrophy patterns in the anterior cingulate and frontoinsula (Perry et al., 2017). Consequently, unlike binary classification, this task exposes the limitations of models lacking strong inductive biases.

In In-Domain (ID), while the hybrid MedViT-3D sets the upper bound, the anatomy-driven SVM establishes a high baseline among standard methods, leveraging precise segmentation priors to disentangle overlapping phenotypes. The 3D ResNet-18 follows closely, confirming that convolutional inductive biases are data-efficient even without explicit segmentation. Conversely, the lack of priors in standard ViT-3D leads to poor convergence. However, the hierarchical Swin-3D DPL variant significantly mitigates this issue, reaching an MCC of 64.84%, thereby narrowing the performance gap with CNNs compared to the vanilla ViT.

The OOD evaluation reveals a shift in the performance hierarchy. While the SVM dominates ID, the 3D ResNet-18 achieves robust generalization performance among non-hybrid architectures, marginally outperforming the SVM. This indicates that end-to-end convolutional features generalize slightly better to site-specific variations than fixed segmentation priors in this 3-class regime. Among Transformers, Swin-DPL remains the most viable option, significantly outperforming the standard Swin-3D and ViT-3D. As observed in the 5-class task, ViT-3D's seemingly favorable ECE is deceptive, resulting from low-confidence predictions rather than accurate calibration.

Table 10: **Baseline performance comparisons for 3-class classification (CN/AD/FTD).** Performance metrics for Convolutional (ResNet, SVM) and Transformer (ViT, Swin) architectures. Results are reported for in-domain (10-fold CV) and out-of-domain (10 models average) settings. **Bold** indicates the best performance per column.
Values: Mean $\pm$ 95% CI ($B = 10,000$). $\uparrow$=higher-is-better, $\downarrow$=lower-is-better.

| Configuration | # Params | ACC $\uparrow$ | MCC $\uparrow$ | PR-AUC $\uparrow$ | Macro-F1 $\uparrow$ | ECE $\downarrow$ | Brier $\downarrow$ |
|---|---|---|---|---|---|---|---|
| In-domain (10-fold CV) | | | | | | | |
| CNNs + SVM | $\approx$270M[*] | $84.25_{\pm1.46}$ | $71.62_{\pm2.56}$ | $85.02_{\pm1.89}$ | $78.77_{\pm2.01}$ | 33.04 | 13.90 |
| ResNet-18 3D | 33.16M | $82.40_{\pm1.50}$ | $68.47_{\pm2.65}$ | $83.43_{\pm1.97}$ | $77.25_{\pm1.98}$ | 28.88 | 13.32 |
| MedViT 3D | 34.99M | $\mathbf{85.92}_{\pm1.38}$ | $\mathbf{74.68}_{\pm2.41}$ | $\mathbf{86.51}_{\pm1.79}$ | $\mathbf{80.77}_{\pm1.92}$ | 32.89 | $\mathbf{13.13}$ |
| ViT-3D | 23.18M | $68.97_{\pm1.83}$ | $43.88_{\pm2.91}$ | $61.81_{\pm2.19}$ | $55.08_{\pm2.26}$ | $\mathbf{21.43}^{\dagger}$ | 16.83 |
| Swin-3D | 29.27M | $75.50_{\pm1.68}$ | $55.53_{\pm2.89}$ | $72.30_{\pm2.31}$ | $65.36_{\pm2.31}$ | 26.02 | 15.65 |
| Swin-3D DPL | 41.02M | $80.54_{\pm1.52}$ | $64.84_{\pm2.64}$ | $79.83_{\pm2.15}$ | $73.18_{\pm2.15}$ | 30.10 | 14.63 |
| Out-of-domain (10 models averaged predictions) | | | | | | | |
| CNNs + SVM | $\approx$270M[*] | $88.52_{\pm1.15}$ | $70.18_{\pm2.87}$ | $83.84_{\pm2.48}$ | $77.31_{\pm2.35}$ | 37.44 | 13.26 |
| ResNet-18 3D | 33.16M | $88.73_{\pm1.15}$ | $71.09_{\pm2.87}$ | $85.88_{\pm2.23}$ | $80.63_{\pm2.17}$ | 36.21 | 12.68 |
| MedViT 3D | 34.99M | $\mathbf{91.23}_{\pm1.05}$ | $\mathbf{76.95}_{\pm2.65}$ | $\mathbf{88.93}_{\pm2.07}$ | $\mathbf{84.98}_{\pm1.98}$ | 38.97 | $\mathbf{12.53}$ |
| ViT-3D | 23.18M | $78.17_{\pm1.49}$ | $45.77_{\pm3.19}$ | $68.49_{\pm2.84}$ | $56.57_{\pm2.85}$ | $\mathbf{30.51}^{\dagger}$ | 15.51 |
| Swin-3D | 29.27M | $82.63_{\pm1.38}$ | $56.76_{\pm3.22}$ | $75.01_{\pm2.80}$ | $70.42_{\pm2.69}$ | 33.90 | 14.82 |
| Swin-3D DPL | 41.02M | $85.23_{\pm1.27}$ | $63.35_{\pm3.03}$ | $81.05_{\pm2.51}$ | $74.46_{\pm2.47}$ | 34.88 | 13.97 |

[*]Includes the parameters of the underlying segmentation backbone, AssemblyNet (Coupé et al., 2020), composed of 125 U-Nets ($\approx$2.17M params each). [†]Low ECE for ViT-3D reflects under-confidence due to poor discrimination, not effective calibration.

Figure 6 profiles the stochastic stability via the nCV. The SVM (purple bars) serves as a stability lower bound (nCV < 0.1), confirming that observed instability in deep models arises from weight optimization rather than aleatoric uncertainty. Standard ViT-3D exhibits substantial volatility (orange bars), with variance spikes indicating sensitivity to initialization. Crucially, the introduction

of deformable patches in Swin-DPL (green bars) acts as a stabilizer, reducing variance to levels approaching the ResNet baseline (blue bars), suggesting that constraining the attention mechanism effectively smooths the optimization landscape.

Moving to the 5-class setting, Figure 7 illustrates how increased task complexity disproportionately affects Transformer stability. ViT-3D nCV spikes in the OOD setting, indicating inconsistent decision boundaries for minority FTD subtypes. The SVM baseline retains a low variance profile, suggesting that predefined U-Net features provide a representation space less susceptible to optimization noise driven by class imbalance. Although Swin-3D DPL improves mean performance over standard Swin, it retains discernible seed-to-seed variability compared to CNN baselines.

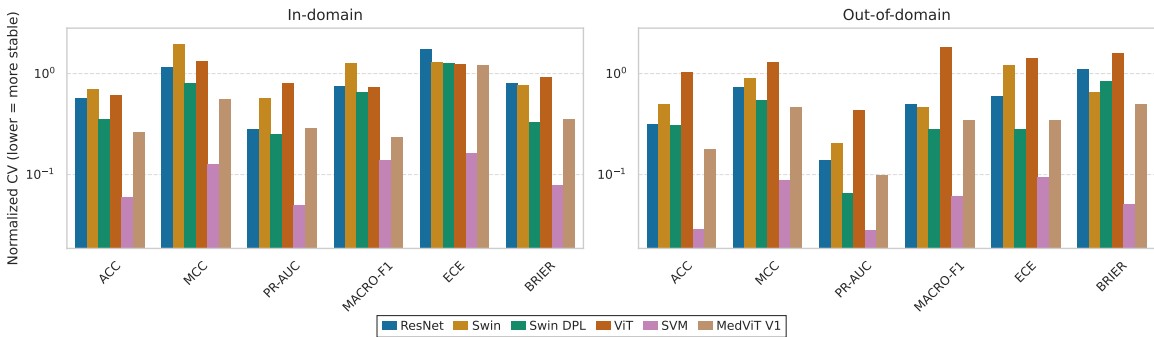

Figure 6: **Inter-seed stability profiling (3-class).** Normalized Coefficient of Variation across 5 random seeds for the 3-class task. Comparison between In-domain (left) and Out-of-domain (right) regimes. Lower nCV values indicate higher stability against initialization noise.

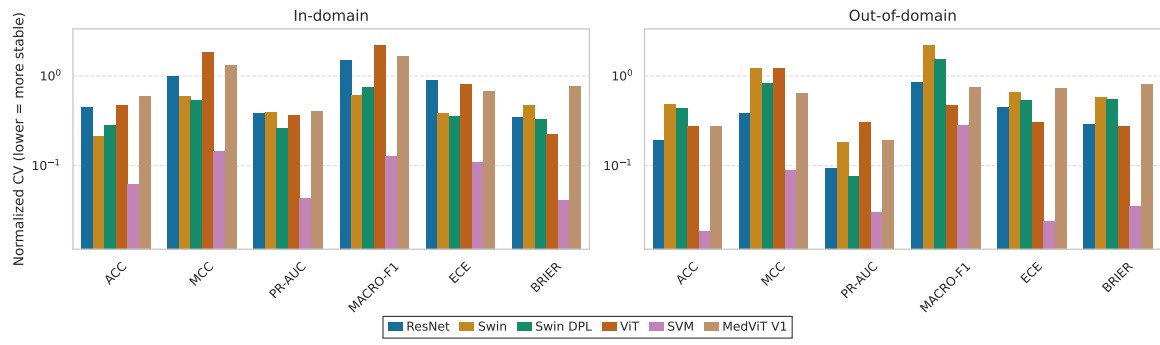

Figure 7: **Inter-seed stability profiling (5-class).** Normalized Coefficient of Variation across 5 random seeds for the differential diagnosis task. Comparison between In-domain (left) and Out-of-domain (right) regimes. Lower nCV values indicate higher stability against initialization noise.

## Appendix F. Extended stability and granular analysis

While the main text reports macro-averaged stability metrics, Figures 8 through 10 present the nCV for the F1-score of each specific class. This stratified analysis indicates that the stability observed in aggregated metrics is predominantly driven by the majority classes (CN, AD). The minority phenotypes, specifically nfvPPA and svPPA, exhibit notably higher variance across seeds (nCV > 0.1, reaching up to 1.2).

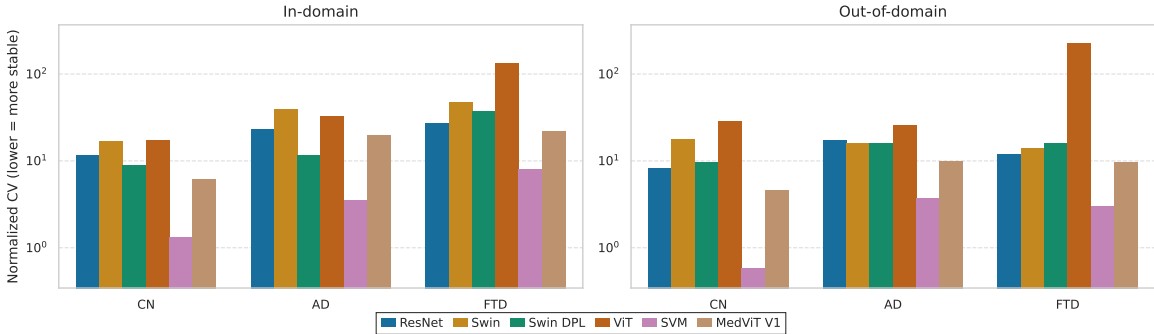

Figure 8: **Per-class F1-score normalized coefficient of variation for the 3-class task (Table 10)** computed across 5 random seeds. The stability profile shows lower variance for majority classes (CN, AD), while the FTD class exhibits higher variability across architectures, particularly for the standard ViT-3D.

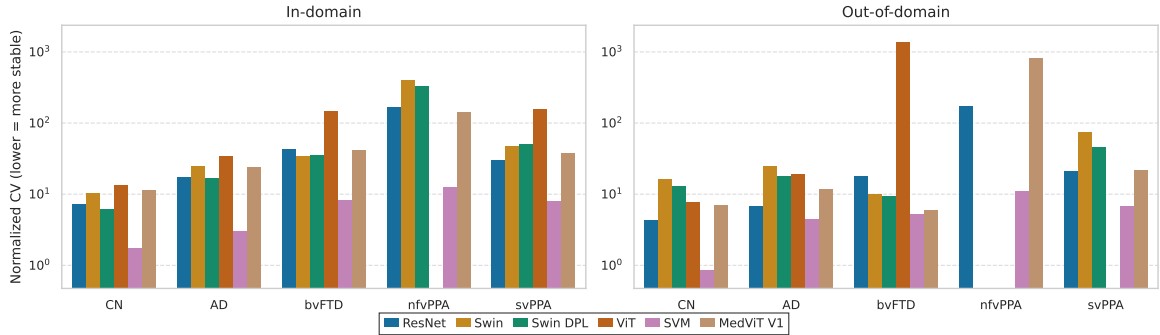

Figure 9: **Per-class F1-score normalized coefficient of variation for the 5-class task (Table 2)** computed across 5 random seeds. The decomposition of the FTD class into subtypes reveals substantial instability for the minority classes (nfvPPA, svPPA), which is masked in macro-averaged metrics.

This breakdown identifies specific limitations of stabilization strategies that are not apparent in global averages. As illustrated in Figure 10, while label smoothing (LS) generally improves calibration metrics, it appears to induce higher variance for the nfvPPA class in the out-of-domain setting (Right panel). This suggests that enforcing soft targets on rare, distinct phenotypes may interfere with robust feature learning under distribution shifts. Conversely, the combined protocol

(`DA+E+LS+BS+M`) effectively suppresses variance for the ID minority classes (Left panel), supporting the synergistic effect of the proposed framework.

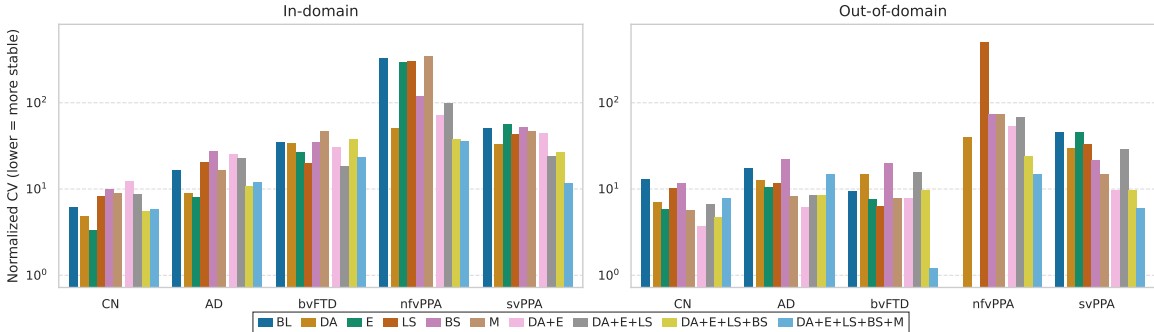

Figure 10: **Per-class F1-score normalized coefficient of variation for training strategies (Table 3)** computed across 5 random seeds. Comparison of stability across diagnosis subtypes for ID and OOD settings. Note the scale difference in variance for minority classes compared to majority classes. While the Baseline (BL) and MixUp (M) show high volatility in-domain for nfvPPA, Label Smoothing (LS) exhibits increased sensitivity for this class in OOD settings.

Figure 11 details the stability profile of the evaluated inference-time strategies. While Test-Time Augmentation (TTA) reduces variance in the in-domain setting, it fails to generalize this benefit to the out-of-domain cohort. Specifically, TTA exhibits higher volatility than the baseline for calibration metrics (ECE, Brier) in OOD. This finding supports the hypothesis that applying geometric transformations (particularly sagittal flipping) to lateralized phenotypes such as nfvPPA and svPPA creates anatomically inconsistent samples that degrade model reliability. Conversely, snapshot ensembling consistently achieves the lowest nCV across all discrimination and reliability metrics, confirming that averaging predictions across validation checkpoints is the most robust strategy to mitigate stochastic optimization variability.

## Appendix G. Generalizability of the stabilization protocol across architectures

To assess whether the proposed stabilization protocol $(DA + E + LS + BS + M)$ benefits other Transformer-based architectures beyond Swin-3D DPL, we performed an additional study on the standard ViT-3D and Swin-3D backbones. Table 11 reports the performance of these architectures when trained with the stabilization protocol presented in Table 3.

The results demonstrate that while our protocol significantly improves ViT-3D performance compared to the baseline reported in Table 2, the non-hierarchical ViT still lags behind Swin-based variants. This confirms that while the proposed optimization strategies effectively reduce variance, hierarchical inductive biases (as found in Swin) remain essential for achieving optimal performance in this setup.

## Appendix H. Computational cost analysis

We quantify the computational overhead of the proposed stabilization protocol in Table 12. The analysis reveals that the fully stabilized protocol (`DA+E+LS+BS+M`) incurs an approximate 4.8×

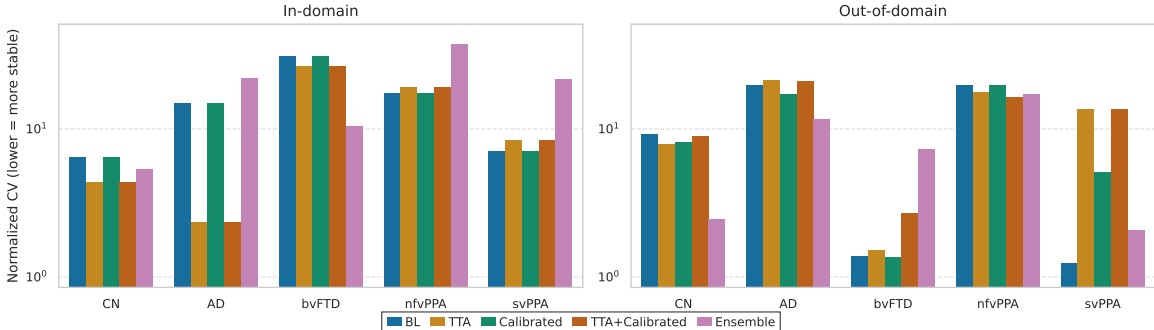

Figure 11: **Normalized coefficient of variation for evaluation strategies.** Comparison of inter-seed stability ($N = 3$ runs) for Baseline (BL), Test-Time Augmentation (TTA), Calibrated models, and Ensemble methods. Ensembling consistently yields the lowest nCV across metrics, indicating that averaging predictions effectively mitigates the variance induced by optimization noise.

Table 11: **Comparison of fully stabilized architectures (ViT vs. Swins).** Performance of distinct backbones trained with the complete optimization protocol (`DA+E+LS+BS+M`). Results are reported for in-domain (ID, 10-fold CV) and out-of-domain (OOD, 10 models average) settings. The stabilization protocol benefits all architectures, but hierarchical models (Swin) retain a significant advantage over vanilla ViT. **Bold** indicates the best performance per column.
Values: Mean $\pm$ 95% CI ($B = 10^4$). ↑=higher-is-better, ↓=lower-is-better.

| | Architecture | Params | ACC ↑ | MCC ↑ | PR-AUC ↑ | Macro-F1 ↑ | ECE ↓ | Brier ↓ |
|---|---|---|---|---|---|---|---|---|
| **ID** | ViT-3D | 23.18M | $69.65 \pm 1.83$ | $44.23 \pm 2.97$ | $43.04 \pm 2.78$ | $43.55 \pm 3.15$ | **40.53**[†] | 13.53 |
| | Swin-3D | 29.27M | $81.12 \pm 1.54$ | $66.66 \pm 2.56$ | $65.85 \pm 3.64$ | $62.60 \pm 3.46$ | 48.18 | 12.05 |
| | Swin-3D DPL | 41.02M | $\mathbf{81.62 \pm 1.54}$ | $\mathbf{67.28 \pm 2.59}$ | $\mathbf{66.43 \pm 3.89}$ | $\mathbf{63.02 \pm 3.71}$ | 48.94 | **12.04** |
| **OOD** | ViT-3D | 23.18M | $84.65 \pm 1.30$ | $61.38 \pm 4.31$ | $64.61 \pm 4.56$ | $63.47 \pm 4.43$ | 56.48 | 13.22 |
| | Swin-3D | 29.27M | $\mathbf{88.48 \pm 1.15}$ | $70.27 \pm 2.87$ | $\mathbf{75.45 \pm 4.48}$ | $70.57 \pm 4.41$ | **55.51** | **11.68** |
| | Swin-3D DPL | 41.02M | $88.41 \pm 1.19$ | $\mathbf{70.38 \pm 2.87}$ | $74.91 \pm 4.90$ | $\mathbf{71.54 \pm 4.52}$ | 55.56 | 11.71 |

[†]Low ECE here reflects under-confidence due to poor discrimination, rather than effective calibration.

increase in training time compared to the baseline (0.93h vs. 4.51h per fold on a V100 GPU). This overhead is primarily driven by the on-the-fly 3D data augmentation and the balanced sampling routine. However, this additional cost must be weighed against the substantial improvements in generalization and robustness: the stabilized model achieves a gain of +15.4 points in OOD Macro-F1 and +6.5 points in OOD MCC compared to the baseline (see Table 3). In the context of medical differential diagnosis, where model failure can have severe consequences, we argue that this trade-off is favorable, as the absolute training duration remains compatible with clinical research workflows ($< 5$ hours per fold).

Table 12: **Computational Analysis (Swin 3D DPL).** Evaluation of training cost and environmental impact across configurations. All models were trained on Jean Zay ($2\times$V100) with 5 repetitions of 10-fold cross-validation (50 runs total). Time is reported per fold (mean $\pm$ std). Total $CO_2$ represents the cumulative footprint of the 50 runs[a].
Abbreviations: BS: Balanced Sampling, M: MixUp, EMA: Exp. Moving Avg, LS: Label Smoothing, DA: 3D Augmentation.

| Configuration | Time / Fold (h) | Steps / Fold | Total Eq. $CO_2$ (kg) |
|---|---|---|---|
| *Single Component Analysis* | | | |
| Baseline | $0.93 \pm 0.07$ | $434 \pm 16$ | $2.40 \pm 0.18$ |
| DA | $2.78 \pm 0.52$ | $1012 \pm 181$ | $7.17 \pm 1.34$ |
| EMA | $0.93 \pm 0.04$ | $436 \pm 14$ | $2.40 \pm 0.10$ |
| LS | $0.95 \pm 0.09$ | $456 \pm 40$ | $2.45 \pm 0.23$ |
| BS | $0.83 \pm 0.04$ | $409 \pm 19$ | $2.14 \pm 0.10$ |
| M | $0.89 \pm 0.05$ | $435 \pm 22$ | $2.30 \pm 0.13$ |
| *Cumulative Complexity* | | | |
| + DA | $2.78 \pm 0.52$ | $1012 \pm 181$ | $7.17 \pm 1.34$ |
| + DA + EMA | $2.70 \pm 0.46$ | $974 \pm 165$ | $6.97 \pm 1.19$ |
| + DA + EMA + LS | $3.05 \pm 0.79$ | $1110 \pm 292$ | $7.87 \pm 2.04$ |
| + DA + EMA + LS + BS | $4.59 \pm 1.72$ | $1624 \pm 509$ | $11.84 \pm 4.44$ |
| + DA + EMA + LS + BS + M | $4.51 \pm 1.37$ | $1648 \pm 496$ | $11.64 \pm 3.53$ |

[a]Based on established factor of $\approx 25.8$ gCO$_2$e/h for Jean Zay V100 using https://labos1point5.org/les-rapports/estimation-empreinte-calcul.

