# OpenReview forum: "On the Stability and Robustness of Vision Transformers for Neurodegenerative Disease Classification"
_MIDL.io/2026/Validation_Papers — MIDL 2026 - Validation Papers Poster_

### Official Review · Reviewer_CvpD · 2025-12-28

**Confidence:** 4
**Preliminary Rating:** 4
**Final Rating:** 4

**Summary:**

This study establishes a comprehensive benchmark centered on Transformer–based models and investigates a protocol that combines data augmentation, architectural constraints, and optimization strategies on multi-site MRI datasets. The authors aim to characterize model stability and uncertainty under increasing task complexity, employing patient-level paired bootstrapping, calibration analysis, paired significance testing, and estimates of the probability of false outperformance.

**Strengths:**

To the best of my knowledge, transformer-based models often struggle when applied to 3D medical imaging data and, in many cases, underperform compared to even some CNN-based architectures. From this perspective, the present study tackles a genuinely challenging and important problem by systematically evaluating Swin Transformer variants in multi-site MRI settings. The paper provides a targeted and valuable comparison, accompanied by an in-depth analysis of different training stabilization techniques, which is highly informative for practitioners working in this domain. Also the inclusion of both IID and OOD scenarios, together with uncertainty-aware metric significantly strengthens the study.

**Weaknesses:**

1. Some of the baseline methods appear somewhat outdated, which may limit the strength of the conclusions when comparing transformer-based models against the broader state of the art.
2. The Alzheimer’s disease classification task seems relatively simple, making it difficult to clearly separate the strengths and weaknesses of different approaches based on performance alone. In several cases, the reported variance is comparable to or even larger than the observed performance improvements, which raises concerns about the practical significance of the numerical differences.
3. In the OOD setting, the calibration results are somewhat puzzling. Specifically, models with lower ECE sometimes exhibit worse classification performance, for example in the case of ViT-3D. This behavior is counterintuitive, as OOD generalization is typically expected to correlate strongly with calibration quality. It would be helpful to examine whether negative log-likelihood (NLL) shows a similar trend, which could clarify whether this phenomenon reflects a broader calibration issue or an artifact of ECE.
4. Although stratification based on age and sex is mentioned, the included datasets do not appear to be explicitly age- and sex-matched. Please report statistical tests assessing group differences in these demographic variables. If significant differences exist, this could have a severe confounding effect on AD classification, potentially undermining the validity of the results. A more robust approach would be to resample matched sub-cohorts to minimize this source of bias.

**Detailed Comments:**

See above

**Justification Of Final Rating:**

Most of my concerns have been adequately addressed. Overall, this is a solid and interesting piece of work. The only remaining comment is the issue of age mismatch in the datasets. I would encourage the authors to explicitly discuss this limitation in the final version, to avoid potential confusion or misinterpretation by future readers and follow-up studies.

**Justification Of The Preliminary Rating:**

There is something unclear about data and leave some room for confusion. Aside from these issues, I do not observe other major weaknesses, and the study remains a meaningful and well-motivated contribution.

**Questions To Address In The Rebuttal:**

see weakness

---

### Official Review · Reviewer_Ab8u · 2026-01-03

**Confidence:** 4
**Preliminary Rating:** 5
**Final Rating:** 5

**Summary:**

This paper systematically investigates the stability, robustness, and reproducibility of Vision Transformer (ViT) models for multiclass neurodegenerative disease classification from structural MRI, with a particular focus on differential diagnosis involving Frontotemporal Dementia subtypes. The authors demonstrate that while Transformers often appear competitive in binary or low-complexity settings, their performance becomes highly seed-dependent and unstable as task complexity, class imbalance, and phenotypic overlap increase. To address this, the paper introduces a composite stabilization protocol combining domain-specific 3D data augmentation, optimization smoothing (e.g., EMA, MixUp), and balanced sampling, and evaluates it through a rigorous multi-seed, uncertainty-aware evaluation framework. Using patient-level paired bootstrapping, calibration analysis, and the Probability of False Outperformance (PFO), the study shows that many reported gains disappear once stochastic variability is properly quantified, highlighting the necessity of uncertainty-aware reporting for reliable medical AI. Overall, the work provides an important methodological contribution by reframing performance evaluation from point estimates to stability- and uncertainty-aware comparisons, with direct implications for clinical deployment.

**Strengths:**

A major strength of this paper is its methodological rigor and its explicit focus on reproducibility and uncertainty, which are critically under-addressed in medical imaging literature. The authors go well beyond standard single-seed evaluations by adopting multi-seed training, paired bootstrapping, Wilcoxon and McNemar tests, and PFO analysis, allowing them to disentangle genuine architectural improvements from optimization noise. This alone makes the paper highly valuable to the community.

The experimental design is also strong: the use of both in-domain (ADNI + ALLFTD) and out-of-domain (NIFD + NACC) cohorts provides a realistic assessment of robustness under distribution shift, while preserving natural class imbalance. The comparison across CNNs, pure Transformers, hybrid architectures, and segmentation-based pipelines is balanced and well-motivated, and the rationale for selecting Swin-3D DPL as the stabilization testbed is clearly articulated.

Importantly, the paper does not oversell Transformers; instead, it provides a nuanced and honest analysis showing when and why ViTs fail, and under which conditions their performance becomes reliable. The emphasis on calibration metrics (ECE, Brier score) alongside discrimination metrics strengthens the clinical relevance. The manuscript is well structured, clearly written, and well situated with respect to prior peer-reviewed work in medical imaging, uncertainty quantification, and optimization stability.

**Weaknesses:**

While the paper is strong overall, several limitations deserve discussion. First, although the stabilization protocol is effective, it is computationally expensive, requiring multi-seed training, extensive augmentation, and ensembling. This raises questions about the practical feasibility of the proposed evaluation and training pipeline for many clinical research groups, which could be discussed more explicitly.

Second, the study is restricted to MRI-only inputs, even though the authors correctly note that differential diagnosis of FTD subtypes often relies on multimodal clinical and neuropsychological information. As a result, some of the observed instability may be inherent to the task formulation rather than the models themselves. While this is acknowledged in the limitations, a clearer distinction between label noise and model instability would strengthen the interpretation.

Third, although several architectural stabilization techniques are evaluated, the paper does not explore self-supervised or large-scale pretraining, which recent peer-reviewed studies suggest can significantly alter the stability regime of Transformers in medical imaging. As a consequence, the conclusions are most directly applicable to from-scratch training scenarios.

Finally, the presentation of results is dense. While appropriate for a full paper, some readers may find it challenging to identify the most actionable takeaways without a concise summary table or decision guide translating the findings into concrete best-practice recommendations.

**Detailed Comments:**

The distinction between data-driven variance (bootstrapped confidence intervals) and optimization-induced instability is insightful and could be emphasized earlier in the paper to better frame the methodology.

The normalized coefficient of variation (nCV) is a useful concept; however, a short intuitive explanation in the main text (not only the appendix) would improve accessibility.

Some figures (e.g., stability and significance matrices) are information-dense; additional annotations or simplified summary plots could improve readability.

**Justification Of Final Rating:**

The authors have addressed the reviewer’s concerns thoughtfully and comprehensively, significantly strengthening the paper. Their rebuttal provides clarity on the computational feasibility of their protocol, differentiating between the expensive evaluation protocol used for quantifying variance and the more practical deployment protocol that relies on techniques like EMA, MixUp, and augmentation. This distinction clarifies the practical applicability of the protocol for clinical research groups, mitigating concerns about the protocol's computational cost.

Computational Feasibility: The authors effectively explain that while the evaluation protocol uses expensive multi-seed training, the goal of their stabilization techniques is to reduce variance without the need for costly multi-seed strategies. This makes their approach much more practical for real-world deployment. The added cost analysis in the appendix further supports this argument, demonstrating that their approach can be applied using a single training run or a small ensemble with high confidence.

Label Noise vs. Model Instability: The authors provide strong evidence that the instability observed with Transformers is optimization-driven rather than label noise-related. The comparison with CNNs, which maintain stable performance despite label noise, reinforces this point. By addressing the distinction between label ambiguity and optimization instability, the authors clarify the root cause of the observed variability and strengthen their argument for the necessity of stabilization techniques in Transformer-based models.

Self-Supervised Learning (SSL) and Foundation Models: The authors have justified their focus on from-scratch training in this study, citing clinical realities and the potential instability of fine-tuning foundation models on small, imbalanced datasets. They also acknowledge that SSL and foundation models represent promising future directions, suggesting that combining their stabilization techniques with SSL initialization will be an important next step.

Multimodal Settings: The authors acknowledge that multimodal integration could exacerbate optimization instability in Transformers, requiring the same rigorous stabilization techniques. They clearly state that adding modalities such as PET and clinical data may improve performance, but it will not automatically resolve optimization brittleness. This nuanced discussion sets the stage for future research in multimodal settings.

Presentation of Results: The addition of Table 6, summarizing actionable takeaways for practitioners, addresses the reviewer’s concern about the density of the results. This table provides a concise decision guide, making the paper more accessible and practical for real-world applications.

In conclusion, the authors have effectively addressed the reviewer’s concerns and made valuable revisions, especially in terms of computational feasibility, model instability, and the integration of stabilization techniques. Their work presents a highly relevant contribution to the field of medical AI, particularly in ensuring reliable and reproducible results. The manuscript is now well-positioned for acceptance.

**Justification Of The Preliminary Rating:**

This paper makes a clear, timely, and methodologically important contribution by demonstrating that many reported gains of Vision Transformers in medical imaging do not survive uncertainty-aware evaluation. While the work does not introduce a novel architecture, its contribution lies in how models should be evaluated, stabilized, and compared, which is arguably more impactful for the field at this stage. The experimental design is thorough, the analysis is careful and honest, and the conclusions are well supported by the data. Minor concerns relate mainly to computational cost and generalization beyond MRI-only, from-scratch training. Overall, the paper meets the bar for acceptance and will be valuable to researchers working on reliable and reproducible medical AI.

**Questions To Address In The Rebuttal:**

How do the authors expect the stability profile of pure Transformers to change under self-supervised or foundation-model-style pretraining on large unlabeled neuroimaging datasets?

Can the authors better disentangle the role of label ambiguity (especially for PPA subtypes) from optimization instability in the observed variance?

Do the authors expect their conclusions to transfer to multimodal settings (MRI + PET + tabular data), where Transformers’ tokenization advantages are more pronounced?

---

### Official Review · Reviewer_DhAb · 2026-01-05

**Confidence:** 4
**Preliminary Rating:** 5
**Final Rating:** 5

**Summary:**

This paper investigates the stability and robustness of Vision Transformers (ViTs) for neurodegenerative disease classification from structural MRI. The authors demonstrate that while binary classification shows moderate stability, performance becomes highly variable in 5-class differential diagnosis. They propose a stabilization protocol combining domain-specific 3D MRI augmentation, optimization smoothing, and balanced sampling. A key contribution is the uncertainty-aware evaluation framework using patient-level paired bootstrapping, calibration analysis, and Probability of False Outperformance. Results show that many apparent performance gains disappear once stochastic variability is properly quantified.

**Strengths:**

1. The focus on differential diagnosis between phenotypically overlapping conditions addresses a genuine clinical challenge where imaging-only diagnosis is inherently difficult.
2. Systematic ablation of training-time and inference-time strategies provides actionable insights for practitioners.
3. Including ECE and Brier score alongside discrimination metrics addresses the probabilistic reliability in clinical decision support.
4. This paper is well-written. It is clear, easy to follow, and well-motivated

**Weaknesses:**

1. The stabilization ablation focuses primarily on Swin-3D DPL, while the paper claims broader applicability to "medical ViTs." Evidence on whether these findings transfer to architecturally distinct transformers (e.g., vanilla ViT) would strengthen this claim.
2. The paper does not compare against other established techniques for stabilizing transformer training, such as gradient clipping strategies, alternative learning rate schedules, or warm-up variations, which could provide additional context for the proposed protocol's effectiveness.
3. The proposed stabilization protocol involves multiple components (heavy augmentation, EMA, MixUp, balanced sampling). The additional computational overhead compared to baseline training is not quantified.

**Detailed Comments:**

N/A. This paper is well-written.

**Justification Of Final Rating:**

Thanks for all the detailed reply and revision. The rebuttal has addressed my concerns. I maintain my initial rating and do think this paper is well-written and have its own contirbution to the field.

**Justification Of The Preliminary Rating:**

This paper makes a valuable contribution by demonstrating that proper uncertainty quantification reveals many reported gains as statistically insignificant. The systematic ablation provides actionable guidance for training transformers on limited medical imaging data. The experimental design is thorough with appropriate multi-cohort evaluation and proper statistical testing.

**Questions To Address In The Rebuttal:**

See Weaknesses for major questions.

---

### Author Rebuttal · Authors · 2026-01-23

**Rebuttal:**

We sincerely thank the reviewers for their constructive feedback, which has significantly strengthened this work. We have updated the manuscript to incorporate the requested analyses with all changes highlighted in cyan. Below, we provide individual responses to the points raised in the reviews.

**Supporting Material:**

/attachment/387e11f9632d8fe67d14038094880a221ce0b10b.pdf

---

### Meta-Review · Area_Chair_gs9f · 2026-02-05

**Recommendation:** Accept (Oral)
**Confidence:** 5

**Metareview:**

This is a methodological and well‑written study. Given the consistent strong‑accept ratings and the clear clinical relevance of the protocol and evaluation framework, I recommend acceptance.

---

### Decision · Program_Chairs · 2026-02-14

Accept (Poster)